# MiCE: Mixture of Contrastive Experts for Unsupervised Image Clustering

**Tsung Wei Tsai, Chongxuan Li, Jun Zhu** [*]
Dept. of Comp. Sci. & Tech., Institute for AI, BNRist Center
Tsinghua-Bosch Joint ML Center, THBI Lab, Tsinghua University, Beijing, 100084 China
{peter83112414,chongxuanli1991}@gmail.com, dcszj@mail.tsinghua.edu.cn

## Abstract

We present Mixture of Contrastive Experts (MiCE), a unified probabilistic clustering framework that simultaneously exploits the discriminative representations learned by contrastive learning and the semantic structures captured by a latent mixture model. Motivated by the mixture of experts, MiCE employs a gating function to partition an unlabeled dataset into subsets according to the latent semantics and multiple experts to discriminate distinct subsets of instances assigned to them in a contrastive learning manner. To solve the nontrivial inference and learning problems caused by the latent variables, we further develop a scalable variant of the Expectation-Maximization (EM) algorithm for MiCE and provide proof of the convergence. Empirically, we evaluate the clustering performance of MiCE on four widely adopted natural image datasets. MiCE achieves significantly better results [1] than various previous methods and a strong contrastive learning baseline.

## 1 Introduction

Unsupervised clustering is a fundamental task that aims to partition data into distinct groups of similar ones without explicit human labels. Deep clustering methods (Xie et al., 2016; Wu et al., 2019) exploit the representations learned by neural networks and have made large progress on high-dimensional data recently. Often, such methods learn the representations for clustering by reconstructing data in a deterministic (Ghasedi Dizaji et al., 2017) or probabilistic manner (Jiang et al., 2016), or maximizing certain mutual information (Hu et al., 2017; Ji et al., 2019) (see Sec. 2 for the related work). Despite the recent advances, the representations learned by existing methods may not be discriminative enough to capture the semantic similarity between images.

The instance discrimination task (Wu et al., 2018; He et al., 2020) in contrastive learning has shown promise in pre-training representations transferable to downstream tasks through fine-tuning. Given that the literature (Shiran & Weinshall, 2019; Niu et al., 2020) shows improved representations can lead to better clustering results, we hypothesize that instance discrimination can improve the performance as well. A straightforward approach is to learn a classical clustering model, e.g. spherical $k$-means (Dhillon & Modha, 2001), directly on the representations pre-trained by the task. Such a two-stage baseline can achieve excellent clustering results (please refer to Tab. 1). However, because of the independence of the two stages, the baseline may not fully explore the semantic structures of the data when learning the representations and lead to a sub-optimal solution for clustering.

To this end, we propose Mixture of Contrastive Experts (MiCE), a unified probabilistic clustering method that utilizes the instance discrimination task as a stepping stone to improve clustering. In particular, to capture the semantic structure explicitly, we formulate a mixture of conditional models by introducing latent variables to represent cluster labels of the images, which is inspired by the mixture of experts (MoE) formulation. In MiCE, each of the conditional models, also called an *expert*, learns to discriminate a subset of instances, while an input-dependent *gating function* partitions the dataset into subsets according to the latent semantics by allocating weights among experts. Further, we develop a scalable variant of the Expectation-Maximization (EM) algorithm (Dempster et al.,

---

[*]Corresponding author.
[1]Code is available at: https://github.com/TsungWeiTsai/MiCE

1977) for the nontrivial inference and learning problems. In the E-step, we obtain the approximate inference of the posterior distribution of the latent variables given the observed data. In the M-step, we maximize the evidence lower bound (ELBO) of the log conditional likelihood with respect to all parameters. Theoretically, we show that the ELBO is bounded and the proposed EM algorithm leads to the convergence of ELBO. Moreover, we carefully discuss the algorithmic relation between MiCE and the two-stage baseline and show that the latter is a special instance of the former in a certain extreme case.

Compared with existing clustering methods, MiCE has the following advantages. (i) **Methodologically unified**: MiCE conjoins the benefits of both the discriminative representations learned by contrastive learning and the semantic structures captured by a latent mixture model within a unified probabilistic framework. (ii) **Free from regularization**: MiCE trained by EM optimizes a single objective function, which does not require auxiliary loss or regularization terms. (iii) **Empirically effective**: Evaluated on four widely adopted natural image datasets, MiCE achieves significantly better results than a strong contrastive baseline and extensive prior clustering methods on several benchmarks without any form of pre-training.

## 2 RELATED WORK

**Deep clustering.** Inspired by the success of deep learning, many researchers propose to learn the representations and cluster assignments simultaneously (Xie et al., 2016; Yang et al., 2016; 2017) based on data reconstruction (Xie et al., 2016; Yang et al., 2017), pairwise relationship among instances (Chang et al., 2017; Haeusser et al., 2018; Wu et al., 2019), multi-task learning (Shiran & Weinshall, 2019; Niu et al., 2020), etc. The joint training framework often ends up optimizing a weighted average of multiple loss functions. However, given that the validation dataset is barely provided, tuning the weights between the losses may be impractical (Ghasedi Dizaji et al., 2017).

Recently, several methods also explore probabilistic modeling, and they introduce latent variables to represent the underlying classes. On one hand, deep generative approaches (Jiang et al., 2016; Dilokthanakul et al., 2016; Chongxuan et al., 2018; Mukherjee et al., 2019; Yang et al., 2019) attempt to capture the data generation process with a mixture of Gaussian prior on latent representations. However, the imposed assumptions can be violated in many cases, and capturing the true data distribution is challenging but may not be helpful to the clustering (Krause et al., 2010). On the other hand, discriminative approaches (Hu et al., 2017; Ji et al., 2019; Darlow & Storkey, 2020) directly model the mapping from the inputs to the cluster labels and maximize a form of mutual information, which often yields superior cluster accuracy. Despite the simplicity, the discriminative approaches discard the instance-specific details that can benefit clustering via improving the representations.

Besides, MIXAE (Zhang et al., 2017), DAMIC (Chazan et al., 2019), and MoE-Sim-VAE (Kopf et al., 2019) combine the mixture of experts (MoE) formulation (Jacobs et al., 1991) with the data reconstruction task. However, either pre-training, regularization, or an extra clustering loss is required.

**Contrastive learning.** To learn discriminative representations, contrastive learning (Wu et al., 2018; Oord et al., 2018; He et al., 2020; Tian et al., 2019; Chen et al., 2020) incorporates various contrastive loss functions with different pretext tasks such as colorization (Zhang et al., 2016), context auto-encoding (Pathak et al., 2016), and instance discrimination (Dosovitskiy et al., 2015; Wu et al., 2018). The pre-trained representations often achieve promising results on downstream tasks, *e.g.*, depth prediction, object detection (Ren et al., 2015; He et al., 2017), and image classification (Kolesnikov et al., 2019), after fine-tuning with human labels. In particular, InstDisc (Wu et al., 2018) learns from instance-level discrimination using NCE (Gutmann & Hyvärinen, 2010), and maintains a memory bank to compute the loss function efficiently. MoCo replaces the memory bank with a queue and maintains an EMA of the student network as the teacher network to encourage consistent representations. A concurrent work called PCL (Li et al., 2020) also explores the semantic structures in contrastive learning. They add an auxiliary cluster-style objective function on top of the MoCo's original objective, which differs from our method significantly. PCL requires an auxiliary $k$-means (Lloyd, 1982) algorithm to obtain the posterior estimates and the prototypes. Moreover, their aim of clustering is to induce transferable embeddings instead of discovering groups of data that correspond to underlying semantic classes.

## 3 PRELIMINARY

We introduce the contrastive learning methods based on the instance discrimination task (Wu et al., 2018; Ye et al., 2019; He et al., 2020; Chen et al., 2020), with a particular focus on the recent state-of-the-art method, MoCo (He et al., 2020). Let $\mathbf{X} = \{\mathbf{x}_n\}_{n=1}^N$ be a set of images without the ground-truth labels, and each of the datapoint $\mathbf{x}_n$ is assigned with a unique surrogate label $y_n \in \{1, 2, ..., N\}$ such that $y_n \neq y_j, \forall j \neq n^2$. To learn representations in an unsupervised manner, instance discrimination considers a discriminative classifier that maps the given image to its surrogate label. Suppose that we have two encoder networks $f_{\boldsymbol{\theta}}$ and $f_{\boldsymbol{\theta}'}$ that generate $\ell_2$-normalized embeddings $\mathbf{v}_{y_n} \in \mathbb{R}^d$ and $\mathbf{f}_n \in \mathbb{R}^d$, respectively, given the image $\mathbf{x}_n$ with the surrogate label $y_n$. We show the parameters of the networks in the subscript, and images are transformed by a stochastic data augmentation module before passing to the networks (please see Appendix D). We can model the probability classifier with:

$$p(\mathbf{Y}|\mathbf{X}) = \prod_{n=1}^N p(y_n|\mathbf{x}_n) = \prod_{n=1}^N \frac{\exp(\mathbf{v}_{y_n}^\top \mathbf{f}_n/\tau)}{\sum_{i=1}^N \exp(\mathbf{v}_i^\top \mathbf{f}_n/\tau)}, \tag{1}$$

where $\tau$ is the temperature hyper-parameter controlling the concentration level (Hinton et al., 2015) [3].

The recent contrastive learning methods mainly differ in: (1) The contrastive loss used to learn the network parameters, including NCE (Wu et al., 2018), InfoNCE (Oord et al., 2018), and the margin loss (Schroff et al., 2015). (2) The choice of the two encoder networks based on deep neural networks (DNNs) in which $\boldsymbol{\theta}'$ can be an identical (Ye et al., 2019; Chen et al., 2020), distinct (Tian et al., 2019), or an exponential moving average (EMA) (He et al., 2020) version of $\boldsymbol{\theta}$.

In particular, MoCo (He et al., 2020) learns by minimizing the InfoNCE loss:

$$\log \frac{\exp\left(\mathbf{v}_{y_n}^\top \mathbf{f}_n/\tau\right)}{\exp\left(\mathbf{v}_{y_n}^\top \mathbf{f}_n/\tau\right) + \sum_{i=1}^\nu \exp\left(\mathbf{q}_i^\top \mathbf{f}_n/\tau\right)}, \tag{2}$$

where $\mathbf{q} \in \mathbb{R}^{\nu \times d}$ is a queue of size $\nu \leq N$ storing previous embeddings from $f_{\boldsymbol{\theta}'}$. While it adopts the EMA approach to avoid rapidly changing embeddings in the queue that adversely impacts the performance (He et al., 2020). For convenience, we refer $f_{\boldsymbol{\theta}}$ and $f_{\boldsymbol{\theta}'}$ as the student and teacher network respectively (Tarvainen & Valpola, 2017; Tsai et al., 2019). In the following, we propose a unified latent mixture model based on contrastive learning to tackle the clustering task.

## 4 MIXTURE OF CONTRASTIVE EXPERTS

Unsupervised clustering aims to partition a dataset $\mathbf{X}$ with $N$ observations into $K$ clusters. We introduce the latent variable $z_n \in \{1, 2, ..., K\}$ to be the cluster label of the image $\mathbf{x}_n$ and naturally extend Eq. (1) to Mixture of Contrastive Experts (MiCE):

$$\begin{aligned} p(\mathbf{Y}, \mathbf{Z}|\mathbf{X}) &= \prod_{n=1}^N \prod_{k=1}^K p(y_n, z_n = k|\mathbf{x}_n)^{\mathbb{1}(z_n=k)} \\ &= \prod_{n=1}^N \prod_{k=1}^K p(z_n = k|\mathbf{x}_n)^{\mathbb{1}(z_n=k)} p(y_n|\mathbf{x}_n, z_n = k)^{\mathbb{1}(z_n=k)}, \end{aligned} \tag{3}$$

where $\mathbb{1}(\cdot)$ is an indicator function. The formulation explicitly introduces a mixture model to capture the latent semantic structures, which is inspired by the mixture of experts (MoE) framework (Jacobs et al., 1991). In Eq. (3), $p(y_n|\mathbf{x}_n, z_n)$ is one of the *experts* that learn to discriminate a subset of instances and $p(z_n|\mathbf{x}_n)$ is a *gating function* that partitions the dataset into subsets according to the latent semantics by routing the given input to one or a few experts. With a divide-and-conquer principle, the experts are often highly specialized in particular images that share similar semantics, which improves the learning efficiency. Notably, MiCE is generic to the choice of the underlying

---

[2]The value of the surrogate label can be regarded as the index of the image.

[3]Due to summation over the entire dataset in the denominator term, it can be computationally prohibitive to get Maximum likelihood estimation (MLE) of the parameters (Ma & Collins, 2018).

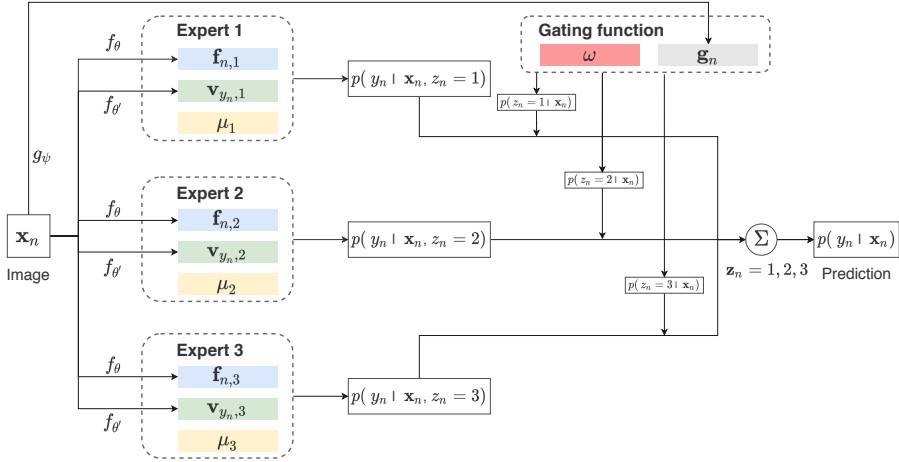

Figure 1: An illustration of MiCE with three experts. Best view in color.

contrastive methods (Wu et al., 2018; He et al., 2020; Chen et al., 2020), while in this paper, we focus on an instance based on MoCo. Also, please see Fig. 1 for an illustration of MiCE with three experts.

In contrast to the original MoE used in the supervised settings (Jacobs et al., 1991), our experts learn from instance-wise discrimination instead of human labels. In addition, both gating and expert parts of MiCE are based on DNNs to fit the high-dimensional data. In the following, we will elaborate on how we parameterize the gating function and the experts to fit the clustering task. For simplicity, we omit the parameters in all probability distributions in this section.

**Gating function.** The gating function organizes the instance discrimination task into $K$ simpler subtasks by weighting the experts based on the semantics of the input image. We define $g_{\boldsymbol{\psi}}$ as an encoder network that outputs an embedding for each input image. We denote the output vector for image $\mathbf{x}_n$ as $\mathbf{g}_n \in \mathbb{R}^d$. The gating function is then parameterized as:

$$p(z_n|\mathbf{x}_n) = \frac{\exp(\boldsymbol{\omega}_{z_n}^\top \mathbf{g}_n/\kappa)}{\sum_{k=1}^K \exp(\boldsymbol{\omega}_k^\top \mathbf{g}_n/\kappa)}, \tag{4}$$

where $\kappa$ is the temperature, and $\boldsymbol{\omega} = \{\boldsymbol{\omega}_k\}_{k=1}^K$ represent the gating prototypes. All prototypes and image embeddings are $\ell_2$-normalized in the $\mathbb{R}^d$ space. Hence, the gating function performs a soft partitioning of the dataset based on the cosine similarity between the image embeddings and the gating prototypes. We can view it as a prototype-based discriminative clustering module, whereas we obtain the cluster labels using posterior inference to consider additional information in the experts.

**Experts.** In MiCE, every expert learns to solve the instance discrimination subtask arranged by the gating function. We define the expert in terms of the unnormalized model $\Phi(\cdot)$ following Wu et al. (2018); He et al. (2020). Therefore, the probability of the image $\mathbf{x}_n$ being recognized as the $y_n$-th one by the $z_n$-th expert is formulated as follows:

$$p(y_n|\mathbf{x}_n, z_n) = \frac{\Phi(\mathbf{x}_n, y_n, z_n)}{Z(\mathbf{x}_n, z_n)}, \tag{5}$$

where $Z(\mathbf{x}_n, z_n) = \sum_{i=1}^N \Phi(\mathbf{x}_n, y_i, z_n)$ is a normalization constant that is often computationally intractable.

Similar to MoCo, we have the student network $f_{\boldsymbol{\theta}}$ that maps the image $\mathbf{x}_n$ into $K$ continuous embeddings $\mathbf{f}_n = \{\mathbf{f}_{n,k}\}_{k=1}^K \in \mathbb{R}^{K \times d}$. Likewise, the teacher network $f_{\boldsymbol{\theta}'}$ outputs $\mathbf{v}_{y_n} = \{\mathbf{v}_{y_n,k}\}_{k=1}^K \in \mathbb{R}^{K \times d}$ given $\mathbf{x}_n$. To be specific, $\mathbf{f}_{n,z_n} \in \mathbb{R}^d$ and $\mathbf{v}_{y_n,z_n} \in \mathbb{R}^d$ are the student embedding and the teacher embedding for images $\mathbf{x}_n$ under the $z_n$-th expert, respectively. We then parameterize the unnormalized model as:

$$\Phi(\mathbf{x}_n, y_n, z_n) = \exp\left(\mathbf{v}_{y_n,z_n}^\top \left(\mathbf{f}_{n,z_n} + \boldsymbol{\mu}_{z_n}\right)/\tau\right), \tag{6}$$

where $\tau$ is the temperature and $\boldsymbol{\mu} = \{\boldsymbol{\mu}_k\}_{k=1}^K$ represent the cluster prototypes for the experts. In Eq. (6), the first instance-wise dot product explores the *instance-level* information to induce discriminative representations within each expert. The second instance-prototype dot product incorporates the *class-level* information into representation learning, encouraging a clear cluster structure around the prototype. Overall, the learned embeddings are therefore encoded with semantic structures while being discriminative enough to represent the instances. Eq. (6) is built upon MoCo with the EMA approach, while in principle, many other potential solutions exist to define the experts, which are left for future studies. Besides, the parameters $\boldsymbol{\theta}$ and $\boldsymbol{\psi}$ are partially shared, please refer to the Appendix D for more details on the architecture.

## 5 INFERENCE AND LEARNING

We first discuss the evidence lower bound (ELBO), the single objective used in MiCE, in Sec. 5.1. Then, we present a scalable variant of the Expectation-Maximization (EM) algorithm (Dempster et al., 1977) to deal with the non-trivial inference and learning of MiCE in Sec. 5.2. Lastly, in Sec. 5.3, we show that a naïve two-stage baseline, in which we run a spherical $k$-means algorithm on the embeddings learned by MoCo, is a special case of MiCE.

### 5.1 EVIDENCE LOWER BOUND (ELBO)

The parameters to update include the parameters $\boldsymbol{\theta}, \boldsymbol{\psi}$ of the student and gating network respectively, and the expert prototypes $\boldsymbol{\mu} = \{\boldsymbol{\mu}\}_{k=1}^K$. The learning objective of MiCE is to maximize the evidence lower bound (ELBO) of the log conditional likelihood of the entire dataset. The ELBO of the datapoint $n$ is given by:

$$\log p(y_n|\mathbf{x}_n) \geq \mathcal{L}(\boldsymbol{\theta}, \boldsymbol{\psi}, \boldsymbol{\mu}; \mathbf{x}_n, y_n)$$
$$:= \mathbb{E}_{q(z_n|\mathbf{x}_n, y_n)}[\log p(y_n|\mathbf{x}_n, z_n; \boldsymbol{\theta}, \boldsymbol{\mu})] - D_{\mathrm{KL}}(q(z_n|\mathbf{x}_n, y_n) \| p(z_n|\mathbf{x}_n; \boldsymbol{\psi})), \quad (7)$$

where $q(z_n|\mathbf{x}_n, y_n)$ is a variational distribution to infer the latent cluster label given the observed data. The first term in Eq. (7) encourages $q(z_n|\mathbf{x}_n, y_n)$ to be high for the experts that are good at discriminating the input images. Intuitively, it can relief the potential *degeneracy* issue (Caron et al., 2018; Ji et al., 2019), where all images are assigned to the same cluster. This is because a degenerated posterior puts the pressure of discriminating all images on a single expert, which may result in a looser ELBO. The second term in Eq. (7) is the Kullback–Leibler divergence between the variational distribution and the distribution defined by the gating function. With this term, the gating function is refined during training and considers the capability of the experts when partitioning data. Notably, MiCE does not rely on auxiliary loss or regularization terms as many prior methods (Haeusser et al., 2018; Shiran & Weinshall, 2019; Wu et al., 2019; Niu et al., 2020) do.

### 5.2 EM ALGORITHM

**E-step.** Inferring the posterior distribution of latent variables given the observations is an important step to apply MiCE to clustering. According to Bayes' theorem, the posterior distribution given the current estimate of the model parameters is:

$$p(z_n|\mathbf{x}_n, y_n; \boldsymbol{\theta}, \boldsymbol{\psi}, \boldsymbol{\mu}) = \frac{p(z_n|\mathbf{x}_n; \boldsymbol{\psi})p(y_n|\mathbf{x}_n, z_n; \boldsymbol{\theta}, \boldsymbol{\mu})}{\sum_{k=1}^K p(k|\mathbf{x}_n; \boldsymbol{\psi})p(y_n|\mathbf{x}_n, k; \boldsymbol{\theta}, \boldsymbol{\mu})}. \quad (8)$$

Comparing with the gating function $p(z_n|\mathbf{x}_n; \boldsymbol{\psi})$, the posterior provides better estimates of the latent variables by incorporating the supplementary information of the experts. However, we cannot tractably compute the posterior distribution because of the normalization constants $Z(\mathbf{x}_n, z_n; \boldsymbol{\theta}, \boldsymbol{\mu})$. In fact, given the image $\mathbf{x}_n$ and the cluster label $z_n$, $Z(\mathbf{x}_n, z_n; \boldsymbol{\theta}, \boldsymbol{\mu})$ sums over the entire dataset, which is prohibitive for large-scale image dataset. We present a simple and analytically tractable estimator to approximate them. Specifically, we maintain a queue $\mathbf{q} \in \mathbb{R}^{\nu \times K \times d}$ that stores $\nu$ previous outputs of the teacher network, following MoCo closely. Formally, the estimator $\hat{Z}(\cdot)$ is:

$$\hat{Z}(\mathbf{x}_n, z_n; \boldsymbol{\theta}, \boldsymbol{\mu}) = \exp\left(\mathbf{v}_{y_n, z_n}^\top (\mathbf{f}_{n, z_n} + \boldsymbol{\mu}_{z_n})/\tau\right) + \sum_{i=1}^\nu \exp\left(\mathbf{q}_{i, z_n}^\top (\mathbf{f}_{n, z_n} + \boldsymbol{\mu}_{z_n})/\tau\right). \quad (9)$$

The estimator is biased, while its bias decreases as $\nu$ increases and we can get a sufficient amount of embeddings from the queue $\mathbf{q}$ efficiently[4]. With Eq. (9), we approximate the posterior as:

$$q(z_n|\mathbf{x}_n, y_n; \boldsymbol{\theta}, \boldsymbol{\psi}, \boldsymbol{\mu}) = \frac{p(z_n|\mathbf{x}_n; \boldsymbol{\psi})\Phi(\mathbf{x}_n, y_n, z_n; \boldsymbol{\theta}, \boldsymbol{\mu})/\hat{Z}(\mathbf{x}_n, z_n; \boldsymbol{\theta}, \boldsymbol{\mu})}{\sum_{k=1}^{K} p(k|\mathbf{x}_n; \boldsymbol{\psi})\Phi(\mathbf{x}_n, y_n, k; \boldsymbol{\theta}, \boldsymbol{\mu})/\hat{Z}(\mathbf{x}_n, k; \boldsymbol{\theta}, \boldsymbol{\mu})}. \quad (10)$$

**M-step.** We leverage the stochastic gradient ascent to optimize ELBO with respect to the network parameters $\boldsymbol{\theta}, \boldsymbol{\psi}$ and the expert prototypes $\boldsymbol{\mu}$. We approximate the normalization constants appear in ELBO in analogy to the E-step, formulated as follows:

$$\widetilde{\mathcal{L}}(\boldsymbol{\theta}, \boldsymbol{\psi}, \boldsymbol{\mu}; \mathbf{x}_n, y_n) = \mathbb{E}_{q(z_n|\mathbf{x}_n, y_n; \boldsymbol{\theta}, \boldsymbol{\psi}, \boldsymbol{\mu})} \left[ \log \frac{\Phi(\mathbf{x}_n, y_n, z_n; \boldsymbol{\theta}, \boldsymbol{\mu})}{\hat{Z}(\mathbf{x}_n, z_n; \boldsymbol{\theta}, \boldsymbol{\mu})} \right]$$
$$- D_{\mathrm{KL}}(q(z_n|\mathbf{x}_n, y_n; \boldsymbol{\theta}, \boldsymbol{\psi}, \boldsymbol{\mu}) \| p(z_n|\mathbf{x}_n; \boldsymbol{\psi})). \quad (11)$$

Sampling a mini-batch $B$ of datapoints, we can construct an efficient stochastic estimator of ELBO over the full dataset to learn $\boldsymbol{\theta}, \boldsymbol{\psi}$ and $\boldsymbol{\mu}$:

$$\mathcal{L}(\boldsymbol{\theta}, \boldsymbol{\psi}, \boldsymbol{\mu}; \mathbf{X}, \mathbf{Y}) \approx \frac{N}{|B|} \sum_{n \in B} \widetilde{\mathcal{L}}(\boldsymbol{\theta}, \boldsymbol{\psi}, \boldsymbol{\mu}; \mathbf{x}_n, y_n). \quad (12)$$

It requires additional care on the update of the prototypes, as discussed in many clustering methods (Sculley, 2010; Xie et al., 2016; Yang et al., 2017; Shiran & Weinshall, 2019). Some of them carefully adjust the learning rate of each prototype separately (Sculley, 2010; Yang et al., 2017), which can be very different from the one used for the network parameters. Since evaluating different learning rate schemes on the validation dataset is often infeasible in unsupervised clustering, we employ alternative strategies which are free from using per-prototype learning rates in MiCE.

As for the expert prototypes, we observe that using only the stochastic update can lead to bad local optima. Therefore, at the end of each training epoch, we apply an additional analytical update derived from the ELBO as follows:

$$\hat{\boldsymbol{\mu}}_k = \sum_{n: \hat{z}_n = k} \mathbf{v}_{y_n, k}, \quad \boldsymbol{\mu}_k = \frac{\hat{\boldsymbol{\mu}}_k}{\|\hat{\boldsymbol{\mu}}_k\|_2}, \quad \forall k, \quad (13)$$

where $\forall n$, $\hat{z}_n = \arg\max_k q(k|\mathbf{x}_n, y_n; \boldsymbol{\theta}, \boldsymbol{\psi}, \boldsymbol{\mu})$ is the hard assignment of the cluster label. Please refer to Appendix A.2 for the detailed derivation. Intuitively, the analytical update in Eq. (13) considers all the teacher embeddings assigned to the $k$-th cluster, instead of only the ones in a mini-batch, to avoid bad local optima.

Beside, we fix the gating prototypes $\boldsymbol{\omega}$ to a set of pre-defined embeddings to stabilize the training process. However, using randomly initialized prototypes may cause unnecessary difficulties in partitioning the dataset if some of them are crowded together. We address the potential issue by using the means of a Max-Mahalanobis distribution (MMD) (Pang et al., 2018) which is a special case of the mixture of Gaussian distribution. The untrainable means in MMD provide the optimal inter-cluster dispersion (Pang et al., 2020) that stabilizes the gating outputs. We provide the algorithm of MMD in Appendix B and a systematical ablation study in Tab. 3 to investigate the effect of the updates on $\boldsymbol{\omega}$ and $\boldsymbol{\mu}$. Lastly, we provide the formal proof on the convergence of the EM algorithm in Appendix A.4.

### 5.3 RELATIONS TO A TWO-STAGE BASELINE

The combination of a contrastive learning method and a clustering method is a natural baseline of MiCE. Our analysis reveals that MiCE is the general form of the two-stage baseline in which we learn the image embeddings with MoCo (He et al., 2020) and subsequently run a spherical $k$-means algorithm (Dhillon & Modha, 2001) to obtain the cluster labels.

On one hand, in the extreme case where $\kappa \to \infty$ (Assumption A3), the student embeddings $\mathbf{f}_{n,k}$ and teacher embeddings $\mathbf{v}_{y_n, k}$ are identical for different $k$ (Assumption A4), and the class-level

---

[4]Even though the bias does not vanish due to the use of queue, we find that the approximation works well empirically.

Table 1: Unsupervised clustering performance of different methods on three datasets. The first sector presents the results from the literature, the later ones display the results of the baseline and the proposed MiCE. In the last two sectors, the bold results indicating the one with the highest values. Methods with the legend† are the ones that required post-processing by $k$-means to obtain the clusters since they do not learn the clustering function directly, except that we use spherical $k$-means for MoCo. We calculate the mean and standard deviation (Std.) of MiCE and MoCo based on five runs.

| Datasets | CIFAR-10 | | | CIFAR-100 | | | STL-10 | | | ImageNet-Dog | | |
|---|---|---|---|---|---|---|---|---|---|---|---|---|
| Methods/Metrics (%) | NMI | ACC | ARI | NMI | ACC | ARI | NMI | ACC | ARI | NMI | ACC | ARI |
| $k$-means (Lloyd, 1982) | 8.7 | 22.9 | 4.9 | 8.40 | 13.0 | 2.8 | 12.5 | 19.2 | 6.1 | 5.5 | 10.5 | 2.0 |
| SC (Zelnik-Manor & Perona, 2004) | 10.3 | 24.7 | 8.5 | 9.0 | 13.6 | 2.2 | 9.8 | 15.9 | 4.8 | 3.8 | 11.1 | 1.3 |
| AE† (Bengio et al., 2006) | 23.9 | 31.4 | 16.9 | 10.0 | 16.5 | 4.8 | 25.0 | 30.3 | 16.1 | 10.4 | 18.5 | 7.3 |
| DAE† (Vincent et al., 2010) | 25.1 | 29.7 | 16.3 | 11.1 | 15.1 | 4.6 | 22.4 | 30.2 | 15.2 | 10.4 | 19.0 | 7.8 |
| SWWAE† (Zhao et al., 2015) | 23.3 | 28.4 | 16.4 | 10.3 | 14.7 | 3.9 | 19.6 | 27.0 | 13.6 | 9.4 | 15.9 | 7.6 |
| GAN† (Radford et al., 2015) | 26.5 | 31.5 | 17.6 | 12.0 | 15.3 | 4.5 | 21.0 | 29.8 | 13.9 | 12.1 | 17.4 | 7.8 |
| VAE† (Kingma & Welling, 2013) | 24.5 | 29.1 | 16.7 | 10.8 | 15.2 | 4.0 | 20.0 | 28.2 | 14.6 | 10.7 | 17.9 | 7.9 |
| JULE (Yang et al., 2016) | 19.2 | 27.2 | 13.8 | 10.3 | 13.7 | 3.3 | 18.2 | 27.7 | 16.4 | 5.4 | 13.8 | 2.8 |
| DEC (Xie et al., 2016) | 25.7 | 30.1 | 16.1 | 13.6 | 18.5 | 5.0 | 27.6 | 35.9 | 18.6 | 12.2 | 19.5 | 7.9 |
| DAC (Chang et al., 2017) | 39.6 | 52.2 | 30.6 | 18.5 | 23.8 | 8.8 | 36.6 | 47.0 | 25.7 | 21.9 | 27.5 | 11.1 |
| DCCM (Wu et al., 2019) | 49.6 | 62.3 | 40.8 | 28.5 | 32.7 | 17.3 | 37.6 | 48.2 | 26.2 | 32.1 | 38.3 | 18.2 |
| IIC (Ji et al., 2019) | - | 61.7 | - | - | 25.7 | - | - | 49.9 | - | - | - | - |
| DHOG (Darlow & Storkey, 2020) | 58.5 | 66.6 | 49.2 | 25.8 | 26.1 | 11.8 | 41.3 | 48.3 | 27.2 | - | - | - |
| AttentionCluster (Niu et al., 2020) | 47.5 | 61.0 | 40.2 | 21.5 | 28.1 | 11.6 | 44.6 | 58.3 | 36.3 | 28.1 | 32.2 | 16.3 |
| MMDC (Shiran & Weinshall, 2019) | 57.2 | 70.0 | - | 25.9 | 31.2 | - | 49.8 | 61.1 | - | - | - | - |
| PICA (Huang et al., 2020) | 59.1 | 69.6 | 51.2 | 31.0 | 33.7 | 17.1 | 61.1 | 71.3 | 53.1 | 35.2 | 35.2 | 20.1 |
| MoCo (Mean)† (He et al., 2020) | 66.0 | 74.7 | 59.3 | 38.8 | 39.5 | 24.0 | 60.5 | 70.7 | 53.0 | 34.2 | 30.8 | 18.4 |
| MoCo (Std.)† (He et al., 2020) | 0.6 | 1.7 | 0.9 | 0.2 | 0.1 | 0.4 | 0.9 | 2.0 | 0.8 | 0.3 | 1.7 | 0.9 |
| MiCE (Mean, **Ours**) | **73.5** | **83.4** | **69.5** | **43.0** | **42.2** | **27.7** | **61.3** | **72.0** | **53.2** | **39.4** | **39.0** | **24.7** |
| MiCE (Std., **Ours**) | 0.2 | 0.2 | 0.3 | 0.5 | 1.4 | 0.4 | 1.2 | 1.8 | 2.4 | 1.8 | 3.0 | 2.4 |
| MoCo (Best)† (He et al., 2020) | 66.9 | 77.6 | 60.8 | 39.0 | 39.7 | 24.2 | 61.5 | 72.8 | 52.4 | 34.7 | 33.8 | 19.7 |
| MiCE (Best, **Ours**) | **73.7** | **83.5** | **69.8** | **43.6** | **44.0** | **28.0** | **63.5** | **75.2** | **57.5** | **42.3** | **43.9** | **28.6** |

information in Eq. (6) is omitted (Assumption A5), we arrive at the same Softmax classifier (Eq. (1)) and the InfoNCE loss (Eq. (2)) used by MoCo as a special case of our method. On the other hand, of particular relevance to the analytical update on expert prototypes (Eq. (13)) is the spherical $k$-means algorithm (Dhillon & Modha, 2001) that leverages the cosine similarity to cluster $\ell_2$-normalized data (Hornik et al., 2012). In addition to Assumptions A3 and A4, if we assume the unnormalized model is perfectly self-normalized (Assumption A2), using the hard assignment to get the cluster labels together with the analytical update turns out to be a single-iteration spherical $k$-means algorithm on the teacher embeddings. Please refer to the Appendix C for a detailed derivation.

The performance of the baseline is limited by the independence of the representation learning stage and the clustering stage. In contrast, MiCE provides a unified framework to align the representation learning and clustering objectives in a principled manner. See a comprehensive comparison in Tab. 1.

## 6 EXPERIMENTS

In this section, we present experimental results to demonstrate the effectiveness of MiCE. We compare MiCE with extensive prior clustering methods and the contrastive learning based two-stage baseline on four widely adopted benchmarking datasets for clustering, including STL-10 (Coates et al., 2011), CIFAR-10 (Krizhevsky et al., 2009), CIFAR-100 (Krizhevsky et al., 2009),

Table 2: Statistics of the datasets.

| Dataset | Images | Clusters | Image Size |
|---|---|---|---|
| CIFAR-10 | 60,000 | 10 | $32 \times 32 \times 3$ |
| CIFAR-100 | 60,000 | 20 | $32 \times 32 \times 3$ |
| STL-10 | 13,000 | 10 | $96 \times 96 \times 3$ |
| ImageNet-Dog | 19,500 | 15 | $96 \times 96 \times 3$ |

and ImageNet-Dog (Chang et al., 2017). The experiment settings follow the literature closely (Chang et al., 2017; Wu et al., 2019; Ji et al., 2019; Shiran & Weinshall, 2019; Darlow & Storkey, 2020) and the numbers of the clusters are known in advance. The statistics of the datasets are summarized in

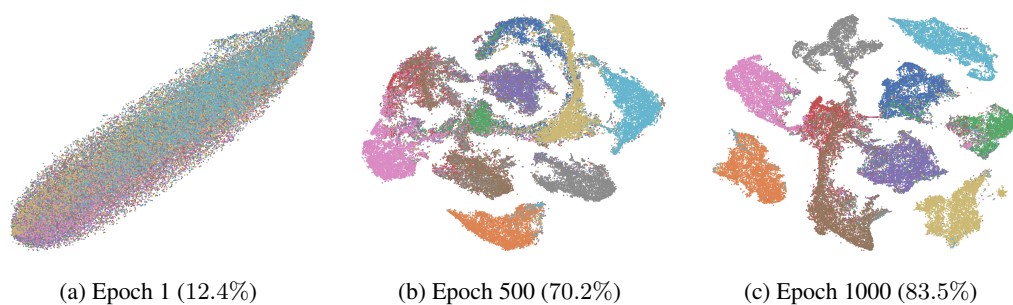

(a) Epoch 1 (12.4%)          (b) Epoch 500 (70.2%)          (c) Epoch 1000 (83.5%)

Figure 2: Visualization of the image embeddings of MiCE on CIFAR-10 with t-SNE. Different colors denote the different ground-truth class labels (unknown to the model). The cluster ACC is shown in the parenthesis. MiCE learns a clear cluster structure that matches the latent semantics well. Best view in color.

Tab. 2. We adopt three common metrics to evaluate the clustering performance, namely normalized mutual information (NMI), cluster accuracy (ACC), and adjusted rand index (ARI). All the metrics are presented in percentage (%). We use a 34-layer ResNet (ResNet-34) (He et al., 2016) as the backbone for MiCE and MoCo following the recent methods (Ji et al., 2019; Shiran & Weinshall, 2019) for fair comparisons. We set both temperatures $\tau$ and $\kappa$ as 1.0, and the batch size as 256. The datasets, network, hyper-parameters, and training settings are discussed detailedly in Appendix D.

## 6.1 MAIN CLUSTERING RESULTS

**Comparison with existing deep clustering methods.** As shown in Tab. 1, MiCE outperforms the previous clustering approaches by a significant margin on all datasets. The comparison highlights the importance of exploring the discriminative representations and the semantic structures of the dataset.

**Comparison with the two-stage baseline.** Compared to the straightforward combination of MoCo and spherical $k$-means, MiCE explores the semantic structures of the dataset that improve the clustering performance. From Tab. 1, we can see that MiCE consistently outperforms the baseline in terms of the mean performance, which agrees with the analysis in Sec. 5.3. Specifically, regarding ACC, we improve upon the strong baseline by 8.7%, 2.7%, and 8.2% on CIFAR-10, CIFAR-100, and ImageNet-Dog, respectively. Taking the measurement variance into consideration, our performance overlaps with MoCo only on STL-10. We conjecture that the small data size may limit the performance as each expert learns from a subset of data. Nevertheless, the comparison manifests the significance of aligning the representation learning and clustering objectives in a unified framework, and we believe that MiCE points out a promising direction for future studies in clustering.

**Visualization of the learned embeddings.** We visualize the image embeddings produced by the gating network using t-SNE (Maaten & Hinton, 2008) in Fig. 2. Different colors denote the different ground-truth class labels. At the beginning, the embeddings from distinct classes are indistinguishable. MiCE progressively refines its estimates and ends up with embeddings that show a clear cluster structure. The learned clusters align with the ground-truth semantics well, which verifies the effectiveness of our method. Additional visualizations and the comparisons with MoCo are in Appendix E.

## 6.2 ABLATION STUDIES

**Simplified model (Tab. 3 (left)).** We investigate the gating function and the unnormalized model to understand the contributions of different components. Using a simpler latent variable model often deteriorates the performance. (1) With a uniform prior, the experts would take extra efforts to become specialized in a set of images with shared semantics. (2 & 3) The teacher embedding $\mathbf{v}_{y_n}$ is pushed to be close to all expert prototypes at the same time. It may be difficult for the simplified expert to encode the latent semantics while being discriminative. (4) The performance drop shows that the class-level information is essential for the image embeddings to capture the semantic structures of the dataset, despite the learned representations are still discriminative between instances. Without the

Table 3: Ablations of MiCE on the probabilistic model (left) and different ways of learning the gating and expert prototypes (right). Each row shows the ACC (%) on CIFAR-100 when applying the single change to MiCE. The assumptions are detailed in the Appendix C.

| | CIFAR-100 | | CIFAR-100 |
|---|---|---|---|
| (1) A3 ( $p(z_n|\mathbf{x}_n) = 1/K$ ) | 40.7 | (a) No analytical update on $\boldsymbol{\mu}$ in Eq. (13) | 21.3 |
| (2) A4 (Single output layer) | 39.3 | (b) No gradient update on $\boldsymbol{\mu}$ | 41.0 |
| (3) A3 + A4 | 33.6 | (c) Initialize $\boldsymbol{\omega}$ with a uniform distribution | 41.0 |
| (4) A5 (Discard class-level information) | 17.0 | (d) Optimize $\boldsymbol{\omega}$ with gradient | 42.0 |
| MiCE (Ours) | 42.2 | MiCE (Ours) | 42.2 |

term, the learned embeddings are mixed up and scattered over the embedding space without a clear cluster structure.

**Prototypes update rules (Tab. 3 (right)).** We also conduct ablation studies to gain insights into the different ways of handling the gating and expert prototypes. We see that (a) without Eq. (13), we may be stuck in bad local optima. As mentioned in Sec. 5.2, a possible reason is that we are using the same learning rate for all network parameters and prototypes (Sculley, 2010; Yang et al., 2017), but tuning separate learning rates for each prototype is impractical for unsupervised clustering. Hence, we derive the analytical update to tackle the issue. As for (b), it shows that the current gradient update rule avoids the potential inconsistency between the expert prototypes and the teacher embeddings during the mini-batch training. Lastly, as discussed in Sec. 5.2, comparing to using (c) uniformly initiated gating prototypes projected onto the unit sphere, utilizing the means of MMD slightly improves performance. This also bypasses the potential learning rate issue that may appear in (d).

# 7 CONCLUSION

We present a principled probabilistic clustering method that conjoins the benefits of the discriminative representations learned by contrastive learning and the semantic structures introduced by the latent mixture model in a unified framework. With a divide-and-conquer principle, MiCE comprises an input-dependent gating function that distributes subtasks to one or a few specialized experts, and $K$ experts that discriminate the subset of images based on instance-level and class-level information. To address the challenging inference and learning problems, we present a scalable variant of Expectation-Maximization (EM) algorithm, which maximizes the ELBO and is free from any other loss or regularization terms. Moreover, we show that MoCo with spherical $k$-means, one of the two-stage baselines, is a special case of MiCE under various assumptions. Empirically, MiCE outperforms extensive prior methods and the strong two-stage baseline by a significant margin on several benchmarking datasets.

For future work, one may explore different learning pretext tasks that potentially fit the clustering task, other than the instance discrimination one. Also, it would be an interesting and important future work to include dataset with a larger amount clusters, such as ImageNet. Besides, being able to obtain semantically meaningful clusters could be beneficial to weakly supervised settings (Zhou, 2018) where quality labels are scarce.

# 8 ACKNOWLEDGEMENTS

The authors would like to thank Tianyu Pang and Zihao Wang for the discussion and the reviewers for the valuable suggestions. This work was supported by NSFC Projects (Nos. 62061136001, 61620106010, 62076145), Beijing NSF Project (No. JQ19016), Beijing Academy of Artificial Intelligence (BAAI), Tsinghua-Huawei Joint Research Program, a grant from Tsinghua Institute for Guo Qiang, Tiangong Institute for Intelligent Computing, and the NVIDIA NVAIL Program with GPU/DGX Acceleration. C. Li was supported by the fellowship of China postdoctoral Science Foundation (2020M680572), and the fellowship of China national postdoctoral program for innovative talents (BX20190172) and Shuimu Tsinghua Scholar.

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

## A  INFERENCE AND LEARNING

### A.1  DERIVATION OF ELBO

$$\log p(\mathbf{Y}|\mathbf{X}; \boldsymbol{\theta}, \boldsymbol{\psi}, \boldsymbol{\mu})$$

$$= \mathbb{E}_{q(\mathbf{Z}|\mathbf{X},\mathbf{Y})} \left[ \log \frac{p(\mathbf{Y}, \mathbf{Z}|\mathbf{X}; \boldsymbol{\theta}, \boldsymbol{\psi}, \boldsymbol{\mu})}{q(\mathbf{Z}|\mathbf{X}, \mathbf{Y})} \right] + D_{\mathrm{KL}}(q(\mathbf{Z}|\mathbf{X}, \mathbf{Y}) \| p(\mathbf{Z}|\mathbf{X}, \mathbf{Y}; \boldsymbol{\theta}, \boldsymbol{\psi}, \boldsymbol{\mu}))$$

$$\geq \mathcal{L}(\boldsymbol{\theta}, \boldsymbol{\psi}, \boldsymbol{\mu}; \mathbf{X}, \mathbf{Y})$$

$$:= \mathbb{E}_{q(\mathbf{Z}|\mathbf{X},\mathbf{Y})} \left[ \log \frac{p(\mathbf{Y}, \mathbf{Z}|\mathbf{X}; \boldsymbol{\theta}, \boldsymbol{\psi}, \boldsymbol{\mu})}{q(\mathbf{Z}|\mathbf{X}, \mathbf{Y})} \right]$$

$$= \mathbb{E}_{q(\mathbf{Z}|\mathbf{X},\mathbf{Y})} \left[ \log p(\mathbf{Y}, \mathbf{Z}|\mathbf{X}; \boldsymbol{\theta}, \boldsymbol{\psi}, \boldsymbol{\mu}) \right] - \mathbb{E}_{q(\mathbf{Z}|\mathbf{X},\mathbf{Y})} \left[ \log q(\mathbf{Z}|\mathbf{X}, \mathbf{Y}) \right]$$

$$= \sum_{n=1}^{N} \sum_{k=1}^{K} q(z_n = k|\mathbf{x}_n, y_n) \left[ \log p(z_n = k|\mathbf{x}_n; \boldsymbol{\psi}) + \log p(y_n|\mathbf{x}_n, z_n = k; \boldsymbol{\theta}, \boldsymbol{\mu}) - \log q(z_n = k|\mathbf{x}_n, y_n) \right].$$

### A.2  DERIVATION OF EQ. (13)

**Assumption A1.** *Under the hard assignment, the variational distribution is given by:*

$$\hat{q}(z_n|\mathbf{x}_n, y_n) = \begin{cases} 1, & \text{if } z_n = \underset{k}{\mathrm{argmax}} \quad q(k|\mathbf{x}_n, y_n), \\ 0, & \text{otherwise.} \end{cases}$$

**Assumption A2.** *For all possible inputs, the unnormalized model $\Phi(\mathbf{x}_n, y_n, z_n)$ is self-normalized similar to (Wu et al., 2018), i.e., $Z(\mathbf{x}_n, z_n)$ equals to a constant c, such that there is no need to approximate the normalization constant as well.*

Here, we derive the analytical update of the expert prototypes based on ELBO. Under the Assumptions A1 and A2, we can update $\boldsymbol{\mu}_k$ by solving the following problem:

$$\boldsymbol{\mu}_k \leftarrow \underset{\boldsymbol{\mu}_k}{\mathrm{argmax}} \, \mathcal{L}(\boldsymbol{\mu}_k; \mathbf{X}, \mathbf{Y})$$

$$= \underset{\boldsymbol{\mu}_k}{\mathrm{argmax}} \sum_{n=1}^{N} \hat{q}(z_n = k|\mathbf{x}_n, y_n) \left[ \log p(k|\mathbf{x}_n) + \log p(y_n|\mathbf{x}_n, k; \boldsymbol{\mu}_k) - \log \hat{q}(z_n = k|\mathbf{x}_n, y_n) \right]$$

$$= \underset{\boldsymbol{\mu}_k}{\mathrm{argmax}} \sum_{n=1}^{N} \hat{q}(z_n = k|\mathbf{x}_n, y_n) \left[ \log p(y_n|\mathbf{x}_n, k; \boldsymbol{\mu}_k) \right]$$

$$= \underset{\boldsymbol{\mu}_k}{\mathrm{argmax}} \sum_{n=1}^{N} \hat{q}(z_n = k|\mathbf{x}_n, y_n) \left[ \log \frac{\Phi(\mathbf{x}_n, y_n, k)}{Z(\mathbf{x}_n, k)} \right]$$

$$= \underset{\boldsymbol{\mu}_k}{\mathrm{argmax}} \sum_{n=1}^{N} \hat{q}(z_n = k|\mathbf{x}_n, y_n) \left[ \log \Phi(\mathbf{x}_n, y_n, k) \right] \qquad \text{(Assumption A2)}$$

$$= \underset{\boldsymbol{\mu}_k}{\mathrm{argmax}} \sum_{n=1}^{N} \hat{q}(z_n = k|\mathbf{x}_n, y_n) \left[ \log \exp \left( \mathbf{v}_{y_n,k}^\top \mathbf{f}_{n,k}/\tau + \mathbf{v}_{y_n,k}^\top \boldsymbol{\mu}_k/\tau \right) \right]$$

$$= \underset{\boldsymbol{\mu}_k}{\mathrm{argmax}} \sum_{n=1}^{N} \hat{q}(z_n = k|\mathbf{x}_n, y_n) \left( \mathbf{v}_{y_n,k}^\top \mathbf{f}_{n,k}/\tau + \mathbf{v}_{y_n,k}^\top \boldsymbol{\mu}_k/\tau \right)$$

$$= \underset{\boldsymbol{\mu}_k}{\mathrm{argmax}} \sum_{n=1}^{N} \hat{q}(z_n = k|\mathbf{x}_n, y_n) \mathbf{v}_{y_n,k}^\top \boldsymbol{\mu}_k/\tau,$$

subjected to the constraint that $\|\boldsymbol{\mu}_k\| = 1$. Introducing a Lagrange multiplier $\lambda$, we then solve the Lagrangian of the objective function:

$$\underset{\boldsymbol{\mu}_k}{\mathrm{argmax}} \quad \lambda(1 - \boldsymbol{\mu}_k^\top \boldsymbol{\mu}_k) + \sum_{n=1}^{N} \hat{q}(z_n = k|\mathbf{x}_n, y_n)\mathbf{v}_{y_n,k}^\top \boldsymbol{\mu}_k/\tau \tag{14}$$

We can get the estimates of $\boldsymbol{\mu}_k$ and $\lambda$ by differentiating the Lagrangian with regards to them and setting the derivatives to zero. To be specific, for $\boldsymbol{\mu}_k$, we have:

$$\nabla_{\boldsymbol{\mu}_k} \quad \lambda(1 - \boldsymbol{\mu}_k^\top \boldsymbol{\mu}_k) + \sum_{n=1}^{N} \hat{q}(z_n = k|\mathbf{x}_n, y_n)\mathbf{v}_{y_n,k}^\top \boldsymbol{\mu}_k/\tau$$

$$= -2\lambda\boldsymbol{\mu}_k + \sum_{n=1}^{N} \hat{q}(z_n = k|\mathbf{x}_n, y_n)\mathbf{v}_{y_n,k}/\tau = 0,$$

such that

$$\boldsymbol{\mu}_k = \frac{\sum_{n=1}^{N} \hat{q}(z_n = k|\mathbf{x}_n, y_n)\mathbf{v}_{y_n,k}}{2\lambda\tau} := \frac{\hat{\boldsymbol{\mu}}_k}{2\lambda\tau}, \tag{15}$$

where we define $\hat{\boldsymbol{\mu}}_k := \sum_{n=1}^{N} \hat{q}(z_n = k|\mathbf{x}_n, y_n)\mathbf{v}_{y_n,k} := \sum_{n:\hat{z}_n=k} \mathbf{v}_{y_n,k}$ for simplicity. Similarly, by differentiating with respect to $\lambda$, we get:

$$\boldsymbol{\mu}_k^\top \boldsymbol{\mu}_k = 1. \tag{16}$$

Substituting Eq. (15) in Eq. (16), we have:

$$1 = \frac{1}{4\tau^2\lambda^2}\hat{\boldsymbol{\mu}}_k^\top \hat{\boldsymbol{\mu}}_k = \frac{1}{4\tau^2\lambda^2}\|\hat{\boldsymbol{\mu}}_k\|^2.$$

Therefore, we obtain the analytical solutions:

$$\lambda = \frac{\|\hat{\boldsymbol{\mu}}_k\|}{2\tau},$$

$$\boldsymbol{\mu}_k = \frac{\hat{\boldsymbol{\mu}}_k}{2\tau\lambda} = \frac{\hat{\boldsymbol{\mu}}_k}{\|\hat{\boldsymbol{\mu}}_k\|}.$$

The last expression is essentially the update for the prototypes of the experts, as presented in Eq. (13) in the main text.

## A.3   PSEUDOCODE OF MiCE

**Algorithm 1:** Pseudocode of MiCE in a PyTorch-like style

```
# encoder_s, encoder_t, encoder_g: student, teacher, and gating network respectively
# K: number of clusters; D: embedding size
# omega: gating prototypes that are fixed to the centers of MMD (KxD)
# mu: expert prototypes (KxD)
# queue: dictionary as a queue of V embeddings (VxKxD)
# m: momentum
# tau, kappa: temperatures for the experts and gating function respectively
teacher.params = student.params  # initialize
mu_hat = zeros((K, D))
for x in loader: # load a mini-batch x with B samples
    x_f = aug(x) # a randomly augmented version
    x_v = aug(x) # another randomly augmented version
    x_g = aug(x) # the other randomly augmented version

    f = encoder_s.forward(x_f) # student embeddings: BxKxD
    v = encoder_t.forward(x_v) # teacher embeddings: BxKxD
    v = v.detach() # no gradient to teacher embeddings
    g = encoder_g.forward(x_g) # gating embeddings: BxD

    v_f = einsum("BKD,BKD->BK", [v, f]) # instance-level information
    v_mu = einsum("BKD,KD->BK", [v, l2normalize(mu)]) # class-level information
    l_pos = (v_f + v_mu)  # positive logits: BxK

    queue_f = einsum("VKD,BKD->BKV", [queue.detach().clone(), f])
    queue_mu = einsum("VKD,KD->KV", [queue, l2normalize(mu)]
    l_neg = queue_f + queue_mu.view(1, K, V) # BxKxV

    experts_logits = cat([l_pos.view(B, K, 1), l_neg], dim=2) / tau # BxKx(1+V)
    experts = Softmax(expert_logits, dim=2)[:, :, 0] # BxK

    gating = Softmax(einsum("KD,BD->BK", [omega.detach().clone(), g]) / kappa) # BxK
    variational_q = einsum("BK,BK->BK", [gating, experts])
    variational_q /= variational_q.sum(dim=1) # BxK

    ELBO = sum(variational_q * (log(gating) + log(experts) - log(variational_q)) / B
    loss = -ELBO

    # SGD update: student and gating networks
    loss.backward()
    update(encoder_s.params)
    update(encoder_g.params)
    # momentum update: teacher network
    encoder_t.params = m*encoder_t.params+(1-m)*encoder_s.params
    # update the queue
    enqueue(queue, v) # enqueue the current mini-batch
    dequeue(queue) # dequeue the earliest mini-batch
    # aggregate the teacher embeddings
    hard_assignments = argmax(variational_q, dim=1) # Bx1
    mu_hat += einsum("BKD,BK->KD", [v, oneHot(hard_assignments)])
# update the expert prototypes
mu = mu_hat
```

einsum: Einstein sum; cat: concatenation; oneHot: One-hot encoding; Softmax: Softmax function; l2normalized: Normalization with $\ell_2$-norm

A.4 CONVERGENCE OF THE PROPOSED EM ALGORITHM

**Theorem 1 (Convergence of MiCE).** *Assume that the log conditional likelihood of the observed data* $\log p(\mathbf{Y}|\mathbf{X}; \boldsymbol{\theta}, \boldsymbol{\psi}, \boldsymbol{\mu})$ *has an upper bound, the approximated ELBO* $\widetilde{\mathcal{L}}(\boldsymbol{\theta}, \boldsymbol{\psi}, \boldsymbol{\mu}; \mathbf{X}, \mathbf{Y})$ *of MiCE is then upper bounded and it will converge to a certain value* $\mathcal{L}^*$ *with the proposed EM algorithm.*

*Proof.* For any given variational distribution $q(z_n|\mathbf{x}_n, y_n)$, the approximated ELBO we optimize in Eq. (11) can be written as the original ELBO in Eq. (7) plus an additional term:

$$\widetilde{\mathcal{L}}(\boldsymbol{\theta}, \boldsymbol{\psi}, \boldsymbol{\mu}; \mathbf{x}_n, y_n)$$

$$= \mathbb{E}_{q(z_n|\mathbf{x}_n, y_n)} \left[ \log \frac{\Phi\left(\mathbf{x}_n, y_n, z_n; \boldsymbol{\theta}, \boldsymbol{\mu}\right)}{\hat{Z}(\mathbf{x}_n, z_n; \boldsymbol{\theta}, \boldsymbol{\mu})} \right] - D_{\mathrm{KL}}(q(z_n|\mathbf{x}_n, y_n) \| p(z_n|\mathbf{x}_n; \boldsymbol{\psi}))$$

$$= \mathbb{E}_{q(z_n|\mathbf{x}_n, y_n)} \left[ \log \frac{\Phi\left(\mathbf{x}_n, y_n, z_n; \boldsymbol{\theta}, \boldsymbol{\mu}\right)}{Z(\mathbf{x}_n, z_n; \boldsymbol{\theta}, \boldsymbol{\mu})} + \log \frac{Z(\mathbf{x}_n, z_n; \boldsymbol{\theta}, \boldsymbol{\mu})}{\hat{Z}(\mathbf{x}_n, z_n; \boldsymbol{\theta}, \boldsymbol{\mu})} \right] - D_{\mathrm{KL}}(q(z_n|\mathbf{x}_n, y_n) \| p(z_n|\mathbf{x}_n; \boldsymbol{\psi}))$$

$$= \mathcal{L}(\boldsymbol{\theta}, \boldsymbol{\psi}, \boldsymbol{\mu}; \mathbf{x}_n, y_n) + \mathbb{E}_{q(z_n|\mathbf{x}_n, y_n)} \left[ \log \frac{Z(\mathbf{x}_n, z_n; \boldsymbol{\theta}, \boldsymbol{\mu})}{\hat{Z}(\mathbf{x}_n, z_n; \boldsymbol{\theta}, \boldsymbol{\mu})} \right].$$

Recall that the normalization constant is

$$Z(\mathbf{x}_n, z_n; \boldsymbol{\theta}, \boldsymbol{\mu}) = \sum_{i=1}^{N} \exp\left(\mathbf{v}_{y_i, z_n}^\top \left(\mathbf{f}_{n, z_n} + \boldsymbol{\mu}_{z_n}\right)/\tau\right),$$

and the approximated normalization constant is

$$\hat{Z}(\mathbf{x}_n, z_n; \boldsymbol{\theta}, \boldsymbol{\mu}) = \sum_{j=1}^{\nu} \exp\left(\mathbf{q}_{j, z_n}^\top \left(\mathbf{f}_{n, z_n} + \boldsymbol{\mu}_{z_n}\right)/\tau\right), \qquad \text{As in Eq. (9)}$$

where $\mathbf{q}$ is a queue that stores the outputs of the teacher network as described in the main text.

If we use the teacher embeddings of the entire dataset as the queue, then the two normalization constants cancel each other and we obtain the original ELBO. In such cases, the convergence can be proved following the standard EM algorithm (Dempster et al., 1977; Wu, 1983).

Otherwise, we can also bound the approximated ELBO. Since that all student and teacher embeddings and expert prototypes are $\ell_2$-normalized, we can bound the log ratio term independent of the choice of the queue:

$$\frac{-2}{\tau} \le \mathbf{v}_{y_i, z_n}^\top \left(\mathbf{f}_{n, z_n} + \boldsymbol{\mu}_{z_n}\right)/\tau \le \frac{2}{\tau},$$

which implies

$$Z(\mathbf{x}_n, z_n; \boldsymbol{\theta}, \boldsymbol{\mu}) \le N \exp(\frac{2}{\tau}),$$

and

$$\hat{Z}(\mathbf{x}_n, z_n; \boldsymbol{\theta}, \boldsymbol{\mu}) \ge \nu \exp(\frac{-2}{\tau}).$$

Given that they are both positive, we can get

$$\log \frac{Z(\mathbf{x}_n, z_n; \boldsymbol{\theta}, \boldsymbol{\mu})}{\hat{Z}(\mathbf{x}_n, z_n; \boldsymbol{\theta}, \boldsymbol{\mu})} \le \log N - \log \nu + \frac{4}{\tau}$$

Therefore, the approximated ELBO is bounded by:

$$\widetilde{\mathcal{L}}(\boldsymbol{\theta}, \boldsymbol{\psi}, \boldsymbol{\mu}; \mathbf{x}_n, y_n) = \mathcal{L}(\boldsymbol{\theta}, \boldsymbol{\psi}, \boldsymbol{\mu}; \mathbf{x}_n, y_n) + \mathbb{E}_{q(z_n|\mathbf{x}_n, y_n)} \left[ \log \frac{Z(\mathbf{x}_n, z_n; \boldsymbol{\theta}, \boldsymbol{\mu})}{\hat{Z}(\mathbf{x}_n, z_n; \boldsymbol{\theta}, \boldsymbol{\mu})} \right]$$

$$\le \log p(y_n|\mathbf{x}_n; \boldsymbol{\theta}, \boldsymbol{\psi}, \boldsymbol{\mu}) + \log N - \log \nu + \frac{4}{\tau}.$$

Similar to the proof of original EM (Wu, 1983), since the approximated ELBO will not decrease in expectation during training, the approximated ELBO of MiCE will converge to a certain value $\mathcal{L}^*$.

$\square$

There are two remarks for Theorem 1: (1) The convergence of the proposed EM relies on similar assumptions of the standard EM algorithm. (2) Due to the extra log ratio term in the approximated ELBO, it will need further analysis to know the exact convergent point comparing to MiCE learning with the standard EM algorithm.

## B   ALGORITHM FOR GENERATING THE CENTERS OF MMD

We provide the algorithm on generating the centers of the Max-Mahalanobis distribution (MMD) in Algorithm 2. The gating prototypes $\boldsymbol{\omega}$ are fix to these centers during training. The algorithm closely follows the one proposed by Pang et al. (2020). For a dataset with $K$ ground-truth classes, we will generate $K$ centers which are all $\ell_2$-normalized in the $\mathbb{R}^d$ space. Please kindly note that the algorithm requires $K \leq (d+1)$ (Pang et al., 2020).

---

**Algorithm 2:** Algorithm to craft the gating prototypes $\boldsymbol{\omega}$ as the centers of MMD

---

**Input:** The dimension of each gating prototypes $d$ and the number of the clusters $K$.
**Initialization:** We initialize the first prototype $\boldsymbol{\omega}_1$ with the first unit basis vector $e_1 \in \mathbb{R}^d$. Rest of the prototypes $\boldsymbol{\omega}_i, i \neq 1$, are initialized with the zero vector $0^d \in \mathbb{R}^d$

1 **for** $i = 2$ **to** $K$ **do**
2     **for** $j = 1$ **to** $i - 1$ **do**
3         $\boldsymbol{\omega}_{ij} = -[1/(K-1) + \boldsymbol{\omega}_i^\top \boldsymbol{\omega}_j]/\boldsymbol{\omega}_{jj}$
4     **end**
5     $\boldsymbol{\omega}_{ii} = \sqrt{1 - \|\boldsymbol{\omega}_i\|_2^2}$
6 **end**
**Return:** The gating prototypes $\boldsymbol{\omega} = \{\boldsymbol{\omega}_i\}_{i=1}^K$.

---

## C   RELATIONS TO THE TWO-STAGE BASELINE

### C.1   CONTRASTIVE LEARNING

**Assumption A3.** *The gating temperature $\kappa \to \infty$ such that for all $k$, the prior*

$$p(z_n|\mathbf{x}_n) = \frac{1}{K}, \quad \forall n.$$

**Assumption A4.** *There is only a single output layer for both student network $f_{\boldsymbol{\theta}}$ and teacher network $f_{\boldsymbol{\theta}'}$ respectively. The resulting embeddings are used across $K$ expert, to be specific, for all possible cases,*

$$\Phi(\mathbf{x}_n, y_n, k) = \exp\left(\mathbf{v}_{y_n}^\top (\mathbf{f}_n + \boldsymbol{\mu}_k)/\tau\right),$$

*where $\mathbf{f}_n = f_{\boldsymbol{\theta}}(\mathbf{x}_n) \in \mathbb{R}^d$, $\mathbf{v}_{y_n}^\top = f_{\boldsymbol{\theta}'}(\mathbf{x}_n) \in \mathbb{R}^d$.*

**Assumption A5.** *The unnormalized model $\Phi(\cdot)$ considers only the instance-level information such that for all possible cases,*

$$\Phi(\mathbf{x}_n, y_n, z_n) = \exp\left(\mathbf{v}_{y_n, z_n}^\top \mathbf{f}_{n, z_n}/\tau\right).$$

**Theorem 2.** *If Assumptions A3-A5 holds, the overall output of MiCE become*

$$p(y_n|\mathbf{x}_n) = \frac{\exp(\mathbf{v}_{y_n}^\top \mathbf{f}_n/\tau)}{\sum_{i=1}^N \exp(\mathbf{v}_i^\top \mathbf{f}_n/\tau)}.$$

*Proof.* Since we the expert

$$p(y_n|\mathbf{x}_n, z_n) = \frac{\Phi(\mathbf{x}_n, y_n, z_n)}{Z(\mathbf{x}_n, z_n)} = \frac{\exp\left(\mathbf{v}_{y_n}^\top (\mathbf{f}_n + \boldsymbol{\mu}_{z_n})/\tau\right)}{\sum_{i=1}^N \exp\left(\mathbf{v}_i^\top (\mathbf{f}_n + \boldsymbol{\mu}_{z_n})/\tau\right)} \qquad \text{(Assumption A4)}$$

$$= \frac{\exp(\mathbf{v}_{y_n}^\top \mathbf{f}_n/\tau)}{\sum_{i=1}^N \exp(\mathbf{v}_i^\top \mathbf{f}_n/\tau)}. \qquad \text{(Assumption A5)}$$

The overall output of MiCE is then

$$p(y_n|\mathbf{x}_n) = \sum_{k=1}^K p(z_n = k|\mathbf{x}_n) p(y_n|\mathbf{x}_n, z_n = k)$$

$$= \sum_{k=1}^K \frac{1}{K} p(y_n|\mathbf{x}_n, z_n = k) \qquad \text{(Assumption A3)}$$

$$= \frac{\exp(\mathbf{v}_{y_n}^\top \mathbf{f}_n/\tau)}{\sum_{i=1}^N \exp(\mathbf{v}_i^\top \mathbf{f}_n/\tau)}.$$

$\square$

The simplified model shown in Theorem 2 is essentially the non-parametric Softmax classifier used by MoCo and InstDisc, which is also related to some recent contrastive learning methods (Ye et al., 2019; Bachman et al., 2019). Note that InstDisc adopts a slightly different implementation in which the teacher network is identical to the student network and the loss function is based on NCE (Gutmann & Hyvärinen, 2010). For detailed comparisons, please refer to He et al. (2020).

**Lemma 1.** *If Assumptions A3-A5 hold, the posterior is uniformly distributed.*

*Proof.*

$$p(z_n|\mathbf{x}_n, y_n) = \frac{p(z_n|\mathbf{x}_n)\Phi(\mathbf{x}_n, y_n, z_n)/Z(\mathbf{x}_n, z_n)}{\sum_{k=1}^K p(k|\mathbf{x}_n)\Phi(\mathbf{x}_n, y_n, k)/Z(\mathbf{x}_n, k)}$$

$$= \frac{\Phi(\mathbf{x}_n, y_n, z_n)/Z(\mathbf{x}_n, z_n)}{\sum_{k=1}^K \Phi(\mathbf{x}_n, y_n, k)/Z(\mathbf{x}_n, k)} \qquad \text{(Assumption A3)}$$

$$= \frac{\frac{\exp(\mathbf{v}_{y_n}^\top \mathbf{f}_n/\tau)}{\sum_{i=1}^N \exp(\mathbf{v}_i^\top \mathbf{f}_n/\tau)}}{\sum_{k=1}^K \frac{\exp(\mathbf{v}_{y_n}^\top \mathbf{f}_n/\tau)}{\sum_{i=1}^N \exp(\mathbf{v}_i^\top \mathbf{f}_n/\tau)}} \qquad \text{(Theorem 2)}$$

$$= \frac{1}{K}.$$

$\square$

**Assumption A6.** *The normalization constant is computationally tractable such that we can have the variational distribution being identical to the posterior distribution.*

**Theorem 3.** *Given that A3-A6 hold,* $p(y_n|\mathbf{x}_n) = \exp(\mathbf{v}_{y_n}^\top \mathbf{f}_n/\tau)/\sum_{i=1}^N \exp(\mathbf{v}_i^\top \mathbf{f}_n/\tau)$, *and the tractable version of ELBO is identical to the form of InfoNCE (Oord et al., 2018) loss used by MoCo.*

*Proof.*

$$\widetilde{\mathcal{L}}(\boldsymbol{\theta}, \boldsymbol{\psi}, \boldsymbol{\mu}; \mathbf{x}_n, y_n) = \mathbb{E}_{q(z_n|\mathbf{x}_n, y_n)}[\log \frac{\Phi(\mathbf{x}_n, y_n, z_n)}{\hat{Z}(\mathbf{x}_n, z_n)}] - D_{\text{KL}}(q(z_n|\mathbf{x}_n, y_n)\|p(z_n|\mathbf{x}_n))$$

$$= \log \frac{\Phi(\mathbf{x}_n, y_n, z_n)}{\hat{Z}(\mathbf{x}_n, z_n)} \qquad \text{(Lemma 1)}$$

$$= \log \frac{\exp\left(\mathbf{v}_{y_n}^\top \mathbf{f}_n/\tau\right)}{\exp\left(\mathbf{v}_{y_n}^\top \mathbf{f}_n/\tau\right) + \sum_{i=1}^\nu \exp\left(\mathbf{q}_i^\top \mathbf{f}_n/\tau\right)}. \qquad \text{(Theorem 2)}$$

$\square$

According to Theorem 2 and 3, we see that MoCo can be viewed as a special case of the proposed MiCE. In other words, they are able to learn the same representations under the same experimental setting if the above assumptions are made.

## C.2 SPHERICAL $k$-MEANS

**Theorem 4.** *If Assumptions A1-A4 hold, the analytical update on the expert prototypes is equivalent to a single-iteration spherical $k$-means algorithm on the teacher embeddings.*

*Proof.* Under the assumptions, the variational distribution reduces to

$$
\begin{aligned}
q(z_n|\mathbf{x}_n, y_n) &= \frac{\Phi\left(\mathbf{x}_n, y_n, z_n\right)}{\sum_{k=1}^{K} \Phi\left(\mathbf{x}_n, y_n, k\right)} && \text{(Assumptions A2 and A3)} \\
&= \frac{\exp\left(\mathbf{v}_{y_{n,z_n}}^{\top} \mathbf{f}_{n,z_n}/\tau + \mathbf{v}_{y_{n,z_n}}^{\top} \boldsymbol{\mu}_{z_n}/\tau\right)}{\sum_{k=1}^{K} \exp\left(\mathbf{v}_{y_{n,k}}^{\top} \mathbf{f}_{n,k}/\tau + \mathbf{v}_{y_{n,k}}^{\top} \boldsymbol{\mu}_k/\tau\right)} \\
&= \frac{\exp\left(\mathbf{v}_{y_n}^{\top} \mathbf{f}_n/\tau + \mathbf{v}_{y_n}^{\top} \boldsymbol{\mu}_{z_n}/\tau\right)}{\sum_{k=1}^{K} \exp\left(\mathbf{v}_{y_n}^{\top} \mathbf{f}_n/\tau + \mathbf{v}_{y_n}^{\top} \boldsymbol{\mu}_k/\tau\right)} && \text{(Assumption A4)} \\
&= \frac{\exp\left(\mathbf{v}_{y_n}^{\top} \boldsymbol{\mu}_{z_n}/\tau\right)}{\sum_{k=1}^{K} \exp\left(\mathbf{v}_{y_n}^{\top} \boldsymbol{\mu}_k/\tau\right)},
\end{aligned}
$$

such that the hard cluster assignment is determined bt the cosine similarity between the teacher embedding and the expert prototypes:

$$
\hat{q}(z_n|\mathbf{x}_n, y_n) = \begin{cases} 1, & \text{if } z_n = \underset{k}{\arg\max} \quad q(k|\mathbf{x}_n, y_n) = \underset{k}{\arg\max} \quad \mathbf{v}_{y_n}^{\top} \boldsymbol{\mu}_k, \\ 0, & \text{otherwise.} \end{cases}
$$

With the simplified expert model, we follow the previous discussion on Eq. (14) and get the Lagrangian of the objective function:

$$
\underset{\boldsymbol{\mu}_k}{\arg\max} \quad \lambda(1 - \boldsymbol{\mu}_k^{\top} \boldsymbol{\mu}_k) + \sum_{n=1}^{N} \hat{q}(z_n = k|\mathbf{x}_n, y_n) \mathbf{v}_{y_n}^{\top} \boldsymbol{\mu}_k/\tau.
$$

The analytical solution is therefore:

$$
\begin{aligned}
\hat{\boldsymbol{\mu}}_k &:= \sum_{n=1}^{N} \hat{q}(z_n = k|\mathbf{x}_n, y_n) \mathbf{v}_{y_n}, \\
\boldsymbol{\mu}_k &= \frac{\hat{\boldsymbol{\mu}}_k}{\|\hat{\boldsymbol{\mu}}_k\|},
\end{aligned}
$$

where the cluster assignment step and prototype update rule are the same as the spherical $k$-means.

$\square$

## D EXPERIEMENT SETTINGS

We mainly compare with the methods that are trained from scratch without using the pre-training model and the experiment settings follow the literature closely (Chang et al., 2017; Wu et al., 2019; Ji et al., 2019; Shiran & Weinshall, 2019; Darlow & Storkey, 2020). For CIFAR-10, CIFAR-100, and STL-10, all the training and test images are jointly utilized, and the 20 superclasses of CIFAR-100 are used instead of the fine labels. The 15 classes of dog images are selected from the ILSVRC2012 1K (Deng et al., 2009) dataset and resized to $96 \times 96 \times 3$ to form the ImageNet-Dog dataset (Chang

et al., 2017; Wu et al., 2019). Note that the numbers of the clusters are known in advance as in Chang et al. (2017); Ji et al. (2019); Wu et al. (2019); Shiran & Weinshall (2019). The statistics of the datasets are summarized in Tab. 2. We adopt three common metrics to evaluate the clustering performance, namely normalized mutual information (NMI), cluster accuracy (ACC), and adjusted rand index (ARI). All the metrics are presented in percentage (%).

Regarding the network architecture, MiCE mainly use a ResNet-34 (He et al., 2016) as the backbone following the recent methods (Ji et al., 2019; Shiran & Weinshall, 2019) for fair comparisons. For the gating network, the output layer is replaced by a single fully connected layer that generates a $\ell_2$-normalized embeddings in $\mathbb{R}^{128}$. As for the student network, it uses the same ResNet-34 backbone as the gating network and includes $K$ more fully connected layers which map the images to $K$ embeddings. Therefore, the parameters of student and gating networks are shared except for the output layers. The teacher network $f_{\theta'}$ is the exponential moving average (EMA) version of the student network $f_{\theta}$, which stabilizes the learning process (He et al., 2020; Tarvainen & Valpola, 2017). The update rule follows $\theta' \leftarrow m\theta' + (1 - m)\theta$ with $m \in [0, 1)$ being the smoothing coefficient. In practice, we let $m = 0.999$ following MoCo. Since the images of CIFAR-10 and CIFAR-100 are smaller than ImageNet images, following (Chen et al., 2020), we replace the first 7x7 Conv of stride 2 with a 3x3 Conv of stride 1 for all experiments on CIFAR-10 and CIFAR-100. The first max-pooling operation is removed as well (Wu et al., 2018; Chen et al., 2020; Ye et al., 2019). Please kindly note that if the first max-pooling operation is not removed, MiCE can still achieve 83.6% ACC on CIFAR-10. For fair comparisons, MoCo also uses a ResNet-34 with a 128-dimensional output and follows the same hyper-parameter settings as MiCE.

As it is often infeasible to tune the hyper-parameters with a validation dataset in real-world clustering tasks (Ghasedi Dizaji et al., 2017), we set both temperatures $\tau$ and $\kappa$ as 1.0. The queue size $\nu$ is set to 12800 for STL-10 because of the smaller data size and 16384 for the other three datasets. The data augmentation follows MoCo closely. Specifically, before passing into any of the embedding networks, images are randomly resized and cropped to the same size, followed by random gray-scale, random color jittering, and random horizontal flip. For a fair comparison, MoCo in the two-stage baseline also uses a ResNet-34 backbone. For all datasets, we use a batch size equals to 256. Note that the data augmentation strategy is critical to contrastive learning methods and MiCE, and we follow the one used by MoCo for fairness.

In terms of the optimization details, we use stochastic gradient descent (SGD) as our optimizer on the negative ELBO. We set the SGD weight decay as 0.0001 and the SGD momentum as 0.9 (He et al., 2020). The learning rate is initiated as 1.0 and is multiplied by 0.1 at three different epochs. For different datasets, the number of training epochs is different to accommodate the data size to have a similar and reasonable training time. For CIFAR-10/100, we train for 1000 epochs in total and multiply the learning rate by 0.1 at 480, 640, and 800 epochs. For STL-10, the total epochs are 6000 and the learning rate is multiplied by 0.1 at 3000, 4000, and 5000 epochs. Lastly, for ImageNet-Dog, the total epochs are 3000 and the learning rate is multiplied by 0.1 at 1500, 2000, and 2500 epochs. Also, the learning rate for expert prototypes is the same as the one for network parameters. All the experiments are trained on a single GPU.

For all experiment settings in the main text, we follow the recent methods (Wu et al., 2019; Ji et al., 2019; Shiran & Weinshall, 2019) closely where the models are trained from scratch. The setting is different from some of the methods including VaDE (Jiang et al., 2016), DGG (Yang et al., 2019), and LTVAE (Li et al., 2019) from two aspects. Firstly, we do not use any form of the pre-trained model. Secondly, we focus on a purely unsupervised setting. In contrast, VaDE (Jiang et al., 2016) and DGG (Yang et al., 2019) use a supervised pre-trained model on ImageNet for STL-10. For fairness, in the original submission, we compare to many previous methods that use the same settings.

## E   ADDITIONAL EXPERIMENTS AND VISUALIZATIONS

**Posterior distribution and cluster predictions.** From the left side of Fig. 3, we can see that in the initial stage (epoch 1), MiCE is yet certain about the cluster labels of any give images. At the end of the training (epoch 1000), the major values of the approximated posterior distribution fall in the $[0, 0.1)$ interval, indicating that the model is confident that images do not belong to those clusters.

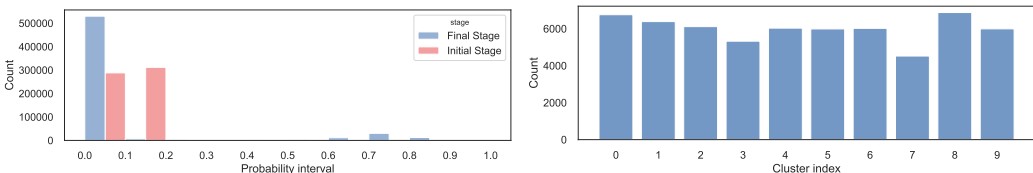

Figure 3: The histogram of (left) approximate posterior distributions of MiCE at the initial and final stage of training and (right) the predicted cluster labels obtained during testing. Here, we train and evaluate the model using CIFAR-10 which has 10 classes and 60,000 images. We divide the probability distribution into 10 discrete bins. Best view in color.

After training, the learned model is able to generate sparse posterior distributions. The predicted cluster labels are also balanced across different clusters, which is shown on the right side of Fig. 3.

**Visualization of embeddings of MiCE and MoCo.** We present the t-SNE visualization of the embeddings learned by MiCE and MoCo in Fig. 4 and Fig. 5 to investigate whether the cluster structure and the latent semantics are captured by the models. The two figures differ in the way we color the datapoints.

In Fig. 4, different colors represent different cluster labels predicted by the clustering methods. We get the predicted cluster labels of MiCE based on the hard assignments using the posterior distributions. The cluster labels for MoCo are the outputs of the spherical $k$-means algorithm. MiCE can learn a distinct structure at the end of the training where each expert would mainly be responsible for one cluster. By comparing Fig. 4 (c) and (f), we can see that the boundaries between the clusters learned by spherical $k$-means do not match the structure learned by MoCo well. The divergence is caused by the independence of representation learning and clustering. MiCE solves the issue with a unified framework.

In Fig. 5, the datapoints are colored according to the ground-truth class labels that are unknown to the models. In Fig. 5(c), the cluster structure is highly aligned with the underlying semantics. Most of the clusters are filled with images from the same classes while having some difficult ones lying mostly on the boundaries. This verifies that the gating network learns to divide the dataset based on the latent semantics and allocates each of the images to one or a few experts. Please kindly note that we use the embeddings from the gating network instead of the student network for simplicity since the embeddings are from the same output head. In contrast, the cluster structure learned by MoCo does not align the semantics well. For all the above t-SNE visualizations, we set the perplexity hyper-parameter to 200.

**Training time.** We present the training time of MiCE and MoCO on CIFAR-10. It takes around 17 and 30 hours to train MoCo and MiCE for 1000 epochs, respectively. For all four datasets, experiments are conducted on a single GPU (NVIDIA GeForce GTX 1080 Ti).

**Extra ablation studies on updating expert prototypes.** A bad specification of the expert prototypes can lead to a bad result if it is not properly handled. Specifically, in Tab. (3) (a), we see that the ACC on CIFAR-10 is only 21.3%. We empirically verify two principled methods that solve the issue: (1) extra end-of-epoch training only on $\mu$ with stochastic gradient ascent or (2) using Eq. (13).

The first method is used as follows. At the end of each epoch, we update $\mu$ while fixing the network parameters until the pre-defined convergence criteria are met. The convergence can be determined based on either the norm of the prototypes or the change of the objective function. To control the training time, we stop the update at the current epoch once we iterate through the entire dataset 20 times. We discover that with additional training, we can achieve 42.3% ACC. The result shows that with a proper update on $\mu$, the proposed model can identify semantic clusters with stochastic gradient ascent alone. However, it requires more than 10 times of training time (around 394 hours).

The above discussions manifest the benefits of using the Eq. (13) which is derived based on the same objective function. With the prototypical update, we can achieve similar results with minimal computational overhead. On average, we can achieve 42.2% ACC on CIFAR-100 as shown in Sec. 6.

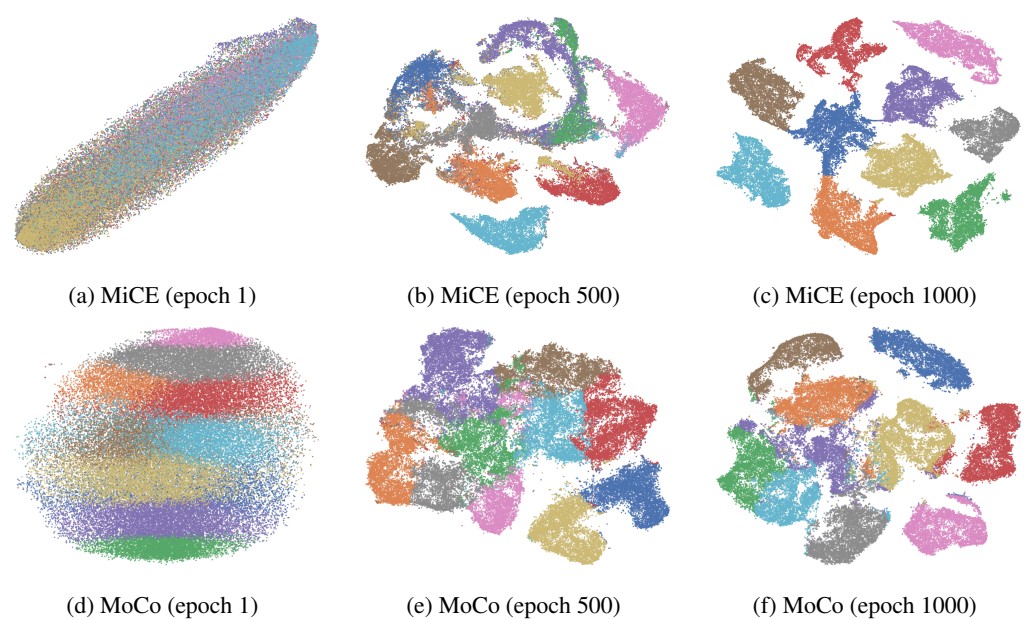

(a) MiCE (epoch 1)      (b) MiCE (epoch 500)      (c) MiCE (epoch 1000)

(d) MoCo (epoch 1)      (e) MoCo (epoch 500)      (f) MoCo (epoch 1000)

Figure 4: Visualization of the image embeddings of MiCE (upper row) and MoCo (lower row) on CIFAR-10 with t-SNE. Different colors correspond to various **cluster labels** obtained based on the posterior distribution (MiCE) or spherical k-means (MoCo). The embeddings of MiCE depict a clear cluster structure, as shown in (c). In contrast, the structure in (f) is ambiguous for a large portion of data and it does not match the cluster labels well.

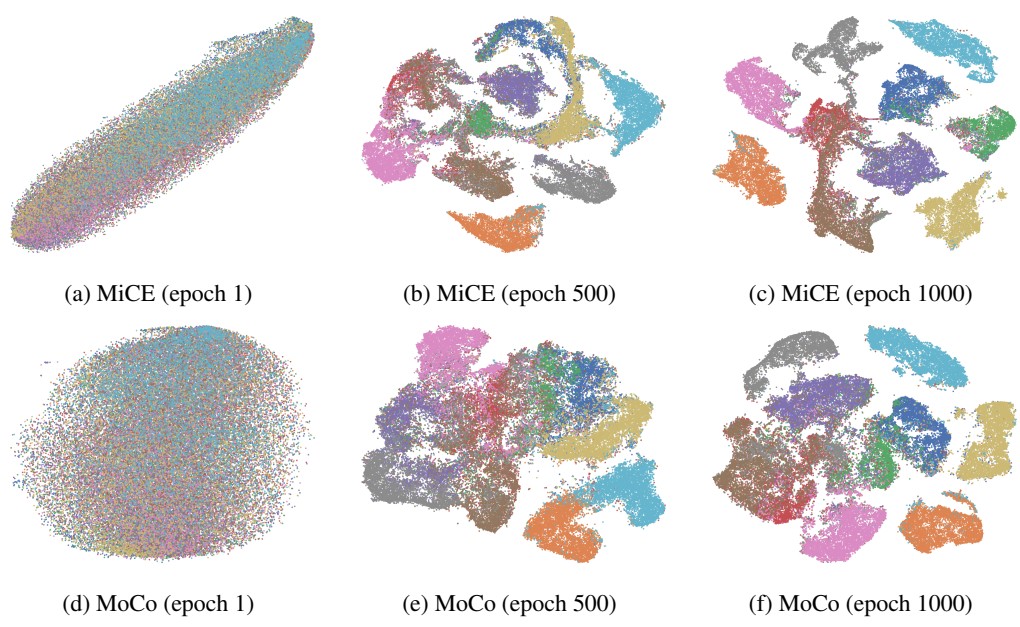

(a) MiCE (epoch 1)      (b) MiCE (epoch 500)      (c) MiCE (epoch 1000)

(d) MoCo (epoch 1)      (e) MoCo (epoch 500)      (f) MoCo (epoch 1000)

Figure 5: Visualization of the image embeddings of MiCE (upper row) and MoCo (lower row) on CIFAR-10 with t-SNE. Different colors denote the different **ground-truth class labels** (unknown to the model). Comparing to MoCo, the clusters learned by MiCE better correspond with the underlying class semantics.

Table 4: Comparing the cluster accuracy ACC (%) of SCAN (Van Gansbeke et al., 2020) and MiCE on CIFAR-10. Following SCAN, we show the data augmentation strategy in the parenthesis if it is different from the one MiCE and MoCo use. "SimCLR" indicates the augmentation strategy used in (Chen et al., 2020), and "RA" is the RandAugment (Cubuk et al., 2020). For a fair comparison, the first sector compares MiCE to SCAN without the self-labeling step, and the second sector compares SCAN with self-labeling to MiCE with pre-training.

| Methods/Dataset | CIFAR-10 |
|---|---|
| SCAN-Loss (SimCLR) | 78.7 |
| SCAN-Loss (RA) | 81.8 |
| MiCE (**Ours**) | **83.4** |
| SCAN-Loss (SimCLR) + Self-Labeling (SimCLR) | 10.0 |
| SCAN-Loss (SimCLR) + Self-Labeling (RA) | 87.4 |
| SCAN-Loss (RA) + Self-Labeling (RA) | 87.6 |
| MiCE pre-training + MiCE (**Ours**) | 89.3 |
| MiCE pre-training + MiCE (RA) (**Ours**) | 88.3 |
| MiCE pre-training + MiCE (SimCLR) (**Ours**) | **90.3** |

**Comparing MiCE to SCAN (Van Gansbeke et al., 2020).** We provide additional experiment results to compare with SCAN (Van Gansbeke et al., 2020) that uses unsupervised pre-training under a comparable experiment setting. As SCAN adopts a three-step training procedure, we think that it will provide additional insights by comparing MiCE to SCAN at the steps after the pre-training step. The detail results on CIFAR-10 are presented in Tab. 4.

Firstly, we focus on SCAN with two steps of training. MiCE outperforms SCAN with two steps of training. SCAN obtains 78.7% and 81.8% on CIFAR-10 with the SimCLR augmentation and RandAugment, respectively. In contrast, MiCE can get 83.4% despite we are using a weaker augmentation strategy following MoCo (He et al., 2020) and InstDisc (Wu et al., 2018).

Since SCAN involves pre-training using SimCLR (Chen et al., 2020), it takes additional advantages when directly comparing to other methods without pre-training. Thus, we fine-tune the MiCE model with the same training protocol described in Appendix D (except the learning rate can be smaller). We discover that MiCE can obtain higher results and outperforms SCAN, as shown in the second sector in Tab. 4. To be specific, in the fine-tuning stage, we load the network parameters from the pre-trained model and a smaller initial learning rate of 0.1 with all the other settings remain the same. We show the augmentation strategy in the parenthesis if we use a different one from the pre-training stage. For the RandAugment (Cubuk et al., 2020), we follow the implementation (and hyper-parameters) based on the public code provided by SCAN. As mentioned in Van Gansbeke et al. (2020), the self-labeling stage requires a shift in the augmentation, otherwise, it will lead to a *degenerated solution*. In contrast, MiCE with pre-training is *not prone to degeneracy can get comparable or better performance than SCAN regardless of the choice of the augmentation strategy*.

## F  ADDITIONAL DISCUSSIONS

**The number of the experts.** In the cases where the number of the experts L differs from the number of ground-truth clusters K, the model will partition the datasets into L subsets instead of K. Even though the number of experts is currently tied with K, it is not a drawback and does not prevent us from applying to the common clustering settings. Also, MiCE does not use additional knowledge comparing to the baseline methods. If the ground-truth K is not known, we may treat K as a hyper-parameter and decide K following the methods described in Smyth (2000); McLachlan & Peel (2004), which is worth investigating in the future.

**Overclustering.** The overclustering technique (Ji et al., 2019) is orthogonal to our methods and can be applied with minor adaptations. However, it may require additional hyper-parameter tuning to ensure overclustering improves the results. From the supplementary file provided by IIC (Ji et al., 2019), we see that the numbers of clusters (for overclustering) are set differently for different datasets.

Despite overclustering is an interesting technique, we would like to highlight the simplicity of the current version of MiCE.

**Similarity between the two set of prototypes.** We do not expect the two prototypes to be similar even though their dimensions are the same. In fact, they have different aims: the gating ones aim to divide the datasets into simpler subtasks for the experts, while the expert ones help the expert to solve the instance discrimination subtasks by introducing the class-level information. Therefore, the derived ELBO objective does not encourage the two sets of prototypes to be similar or maintaining a clear correspondence between them.

Empirically, we calculate the cosine similarity between any pair of $\mu$ and $\omega$. The absolute values we get are less than 0.25, which showed that they are not similar. If we force them to be similar during training, the performance may be negatively affected due to the lack of flexibility.

