# OpenReview forum: "MiCE: Mixture of Contrastive Experts for Unsupervised Image Clustering"
_ICLR.cc/2021/Conference — ICLR 2021 Poster_

### Official Review · AnonReviewer1 · 2020-10-23
**An extension of the MoCo idea in the form of a mixture of MoCo experts for image clustering**

**Rating:** 6
**Confidence:** 5

**Review:**

The paper presents an image clustering methodology based on Mixture of Experts (MoE) for image clustering.
Although MoE has been proposed for supervised learning problems, the authors exploit the instance discrimination framwork to apply the MoE idea for image clustering.

This is a novel aspect of the proposed method.

The MoCo framework (unsupervised) for contrastive learning of image representations is employed to define a mixture of MoCo experts model where each expert additionally includes a cluster prototype vector to facilitate clustering.

This unified approach for simultaneous MoCo-based representation learning and clustering seems to provide better results that the two-stage approach of first applying MoCo and then using k-means clustering on the obtained representations.

A probabilistic formulation of the method is presented, along with a training approach based on EM algorithm for likelihood maximization.

There are several concerns related to presentation and clarity.

Comments to be addressed:
1) It would be easier to understand the contribution of the paper, if the MoCo approach were initially described and then the proposed method was presented as a mixture of MoCo experts. The paper in its current form (section 3) is difficult to follow, since several MoCo ideas are mentioned (eg. student and teacher network,  EMA, etc) without been intuitively explained.
2) In section 3 that describes the method, there is no reference about image augmentation, although it is a critical aspect of the approach. Use of image augmentation is only mentioned at the end of the Appendix.
3) A pseudocode descibing the exact steps of the proposed method is imperative.
4) Due to some approximations made, is it possible to prove convergence of the proposed EM procedure?
5) Gating prototypes \omega remain fixed during training. It is important to provide more details on the MMD method used to specify them.
6) It seems strange that, while \omega are specified using embeddings from the the initial network g(x), they are not involved during training.
7) A bad specification of \omega is expected to have strong negative influence on the results that cannot be recovered.
8) What is the size of minibatch B? (eq. (10)).

---

> ### Author Response · Authors · 2020-11-18
> **Response to Reviewer#1 (1/2)**
>
> We thank the reviewer for the constructive comments and acknowledgment of our novelty. We updated the submission accordingly. In particular, we formally proved the convergence and provided extra pseudocodes and experiments for clarity. Please kindly find the detailed responses below.
>
>
> #### Q1: To introduce MoCo first
> Thank you for the concrete advice. Following the reviewer’s suggestion, we added a preliminary section to introduce contrastive learning, especially MoCo, before introducing MiCE in the revised version to make the paper easier to follow. In the preliminary section, we explained the use of the student and teacher networks, EMA, and data augmentation.
>
> Currently, our writing seems to tie up MiCE and MoCo [2], while we think MiCE is a general framework that can construct the expert model based on different contrastive learning methods such as InstDisc [1] and SimCLR [3] (with some minor adaptations needed). We have also clarified it in Sec.3&4 of the revised version.
>
>
> #### Q2: Reference to data augmentation
> Thank you for pointing out. We updated the submission to mention data augmentation in the preliminary and Sec.6 of the revised version.  For your convenience, we use the standard augmentation as in MoCo [2].
>
>
> #### Q3: Pseudocode
> Thank you for the advice. We added a Pytorch-like pseudocode in Appendix A.3 to show the implementation and exact steps of the proposed method, which follows MoCo [2].
>
>
> #### Q4: Convergence of EM
>
> Thank you for your concern. We formally proved the convergence of MiCE in Appendix A.4. The proof spirit is highly similar to the original EM’s.
>
> In particular, we first rewrite the approximate ELBO used in MiCE as a sum of the original ELBO and a log-ratio between two normalization constants. We found that the log ratio has a constant upper bound. Therefore, by assuming that the log conditional likelihood of the incomplete data has an upper bound, similarly to the original proof in EM, we can upper bound the approximate ELBO. Since the approximated ELBO will not decrease in expectation, MiCE is convergent in expectation as well.

---

> > ### Author Response · Authors · 2020-11-18
> > **Response to Reviewer#1 (2/2)**
> >
> > #### Q5: More details on MMD
> > Thank you for the advice. We have added the algorithm to generate the centers of MMD in Appendix B of the revision. The provided algorithm is conducted before training MiCE to prepare the $\boldsymbol{\omega}$. The centers we used are also provided in the original supplementary file we submitted.
> >
> >
> > #### Q6: It seems strange that, while $\boldsymbol{\omega}$ are specified using embeddings from the initial network g(x), they are not involved during training.
> > There is a potential misunderstanding. The gating prototypes $\boldsymbol{\omega}$ are not specified using the embeddings from the gating network. We have another independent process to generate the means of MMD before training the model. Please kindly refer to Appendix B for the specific algorithm in the revision. Thank you for your concern.
> >
> >
> > #### Q7: A bad specification of $\boldsymbol{\omega}$ is expected to have a strong negative influence on the results that cannot be recovered.
> > Thank you for your concern. This might be a potential misunderstanding. In Tab. 3 (right), we show that MiCE is not sensitive to the different treatments of the gating prototypes $\boldsymbol{\omega}$. As shown in Tab. 3 (b, c, d), they all can get a decent performance on CIFAR-100, while using MMD provides a slightly better result.
> >
> > According to the reviewer’s comment, we suppose that the reviewer is referring to the result of the expert prototypes $\boldsymbol{\mu}$ that is shown in Tab. 3 (a). The bad specification of the expert prototypes can lead to bad results if **it is not properly handled**. We empirically verify *two principled methods that solve the issue*: (1) extra end-of-epoch training only on $\boldsymbol{\mu}$  with stochastic gradient ascent or (2) using Eq. (11). The first method would require a much longer training time, therefore, we are currently using Eq. (11) to prevent the bad specification of $\boldsymbol{\mu}$ during training. We have updated the submission to include the discussion and details in Appendix E for your reference.
> >
> >
> > #### Q8: The size of the mini-batch B
> > Thank you for your reminder. The batch size is set to 256 for all experiments on the four datasets. We updated the submission to include the number in Sec.6.
> >
> >
> >
> >
> > *Lastly, **we would like to express our sincere appreciation for the insightful comments the reviewer provided again**, which indeed help us to improve the submission.*
> >
> >
> >
> >
> >
> > ========================================================================================
> >
> >
> > [1] Zhirong Wu, Yuanjun Xiong, Stella X Yu, and Dahua Lin. Unsupervised feature learning via nonparametric
> > instance discrimination. In Proceedings of the IEEE Conference on Computer Vision
> > and Pattern Recognition, pp. 3733–3742, 2018.
> >
> > [2] Kaiming He, Haoqi Fan, Yuxin Wu, Saining Xie, and Ross Girshick. Momentum contrast for unsupervised visual representation learning. arXiv preprint arXiv:1911.05722, 2019.
> >
> > [3] Ting Chen, Simon Kornblith, Mohammad Norouzi, and Geoffrey Hinton. A simple framework for contrastive learning of visual representations. arXiv preprint arXiv:2002.05709, 2020.
> >
> > [4] Tianyu Pang, Kun Xu, Yinpeng Dong, Chao Du, Ning Chen, and Jun Zhu. Rethinking softmax cross-entropy loss for adversarial robustness. arXiv preprint arXiv:1905.10626, 2019.

---

### Official Review · AnonReviewer4 · 2020-10-28

**Rating:** 8
**Confidence:** 3

**Review:**

Summary and Contributions: Inspired by the mixture of experts, authors propose an image clustering algorithm using a mixture of contrastive experts where, each of the conditional models is an expert in discriminating a subset of instances based on contrastive learning.  To this end they  use a gating function to partition an unlabeled dataset into subsets according to the latent semantics and discriminative distinct, where the gating function performs a soft partitioning of the dataset
based on the cosine similarity between the image embeddings and the gating prototypes. The authors carry out experiments on four widely adopted natural image datasets to evaluate the performance of the method in these tasks and compare it to competing methods and baselines.


Correctness and Clarity: The paper is well-written, with informative figures and tables. The paper presents the idea in a clear and straight-forward manner, and is solidly built on top of the current literature. Authors convincingly tested the method with multiple SOTA and baseline and the results look correct to me.

Reproducibility: The details of the experiments, implementation, and the public datasets are included in the paper. Thanks also for sharing the code.

Additional Feedback and Suggestions:  Since the goal of the paper is image clustering, providing some visual results is appreciated. Also, I am curious to see the performance of the method when we have large number of clusters in our dataset e.g. ImageNet.

Decision: The idea of using  a scalable variant of the Expectation-Maximization (EM) algorithm to help with the nontrivial inference and learning problems caused by the latent variables seems interesting to me.  And overall, the technical novelty together with the fine evaluation are good enough for ICLR, in my opinion.

---

> ### Author Response · Authors · 2020-11-18
> **Response to Reviewer#4**
>
> We thank the reviewer for the positive comments and acknowledgment of our contribution. We revised the submission by providing extra experiments, visualizations, and pseudocodes in both the main text and the appendix. Please kindly find the detailed responses below.
>
>
>
> #### Q1: More visual results
> Thank you for the advice. Following your suggestion, we included several new figures and visualizations to support our ideas in the revised version. For instance, we show the histogram of the posterior estimates of the dataset at the different training epochs. Also, we visualize the image embeddings and the prototypes with t-SNE to investigate whether MiCE capture the cluster structure that matches the latent semantics.
>
> Please kindly refer to Sec.6.1 and Appendix E of the revision for the results.
>
>
>
> #### Q2: The performance on datasets with a larger amount of clusters
> Thank you for your suggestion. We hypothesize that MiCE can perform well on datasets with a larger amount of clusters according to our current empirical results and analysis. This is a very interesting and important future work. We added the discussion in Sec. 7.
>
>
>
> *We would like to express our sincere gratitude and appreciation for the constructive and positive comments again, which help us to improve the quality of the paper.*

---

### Official Review · AnonReviewer2 · 2020-10-29
**Novel methodology to solve a clustering problem with scope of improvement in the paper**

**Rating:** 6
**Confidence:** 5

**Review:**

Summary: Authors present “mixture of experts” type of method to solve a clustering with unsupervised learning problem. Method is called as Mixture of Contrastive Experts (MiCE) which uses contrastive learning as a base module and combines it with latent mixture models. Authors develop a scalable algorithm for MiCE and empirically evaluate the proposed method for image clustering.

Recommendation: I am tending towards accepting the paper (rating 6). Reason for the acceptance is the novel method supported by empirical evidence. Reason for score not being too high are some weaknesses mentioned in the details later.
Strengths:
1) Authors address an image clustering algorithm given number of clusters. Submission is clear, technically correct and present novel findings.
2) The proposed approach is well motivated.

Weakness/Questions:
1) All recent papers have ImageNet-10 as one of the five common datasets [1-2]. Why was it omitted in the current paper?
2) It looks like method is very tied up with MoCo and developed on top of it. Is there an easy/quick way to use other backbones like SimCLR instead of MoCo and still preserve all the steps in the method?
3) Why were images in ImageNet-Dog resized to 96 x 96 x3?
4) Almost all prior methods and proposed method MiCE assume that a number of clusters are known (which shouldn’t ideally be the case). But it looks like the proposed method MiCE uses the information in a better way by assuming number of experts = number of clusters. Can one use more or less number of experts (than number of clusters K) and still partition the sample data into K clusters? Can one easily use over clustering as presented in IIC?
5) How does the proposed method MiCE fare in terms of computation complexity when compared to MoCo?
6) How do \mu, w (the expert and gating prototypes) differ during the training? Since the prediction value is the sum of the expert probability weighted by the gating function, one would expect the gating prototypes and expert prototypes to be similar? Is this true? How is consistency maintained (is there a clear correspondence, prototype 1 in expert matches to prototype 2 in gating function)?
7) Can authors elaborate more about the usage of Max-Mahalanobis distribution (MMD) and how exactly does it solve the issue of “unnecessary difficulties in partitioning the dataset if some of them are crowded together.”
8) “Since the images of CIFAR-10 and CIFAR-100 are smaller than ImageNet images, following (Chen et al., 2020), we replace the first 7x7 Conv of stride 2 with a 3x3 Conv of stride 1 for all experiments on CIFAR-10 and CIFAR-100. The first max-pooling operation is removed as well. For fair comparisons”: Is removing  first max-pooling operation standard practice? Is there any performance loss when max pooling is not removed?

Minor:
1) What does NMI and ARI of > 1 mean? Don’t they have to be in the range of [0,1]?


[1] Huang, Jiabo, Shaogang Gong, and Xiatian Zhu. "Deep Semantic Clustering by Partition Confidence Maximisation." In Proceedings of the IEEE/CVF Conference on Computer Vision and Pattern Recognition, pp. 8849-8858. 2020.
[2] Wu, Jianlong, Keyu Long, Fei Wang, Chen Qian, Cheng Li, Zhouchen Lin, and Hongbin Zha. "Deep comprehensive correlation mining for image clustering." In Proceedings of the IEEE International Conference on Computer Vision, pp. 8150-8159. 2019.

---

> ### Author Response · Authors · 2020-11-18
> **Response to Reviewer#2 (1/3)**
>
> We thank the reviewer for the positive comments and acknowledgment of our novelty. In the revision, we addressed all comments. In particular, we clarified the experiment settings and included the algorithm on the usage of MMD and extra experiments. Please kindly find the detailed responses below.
>
>
> #### Q1: Results on ImageNet-10
> We have started running experiments on the ImageNet-10. Given the limited time of the response period, we plan to include the results in the camera-ready version if possible. Besides, ImageNet-Dog results in the submission can also verify the effectiveness of MiCE on complex datasets.
>
> Also, we added the results from the paper mentioned in the comment with the title “Deep Semantic Clustering by Partition Confidence Maximisation” [8] as one of the baseline methods to Tab.1 of the revised version.
>
>
> #### Q2: The use of other contrastive learning backbones
> Thank you for your concern. The proposed framework is general and can be derived based on different contrastive methods. Indeed, for any contrastive learning methods that can define the model $p(Y|X)$, we can formulate the mixture model by introducing the latent variables as the cluster labels. Incorporating other possible contrastive learning backbones like SimCLR would be interesting future work.
>
> As the current writing seems to be tied up with MoCo, we added a preliminary section (Sec.3 in the revised version) to introduce the contrastive learning before introducing MiCE and clarify their relations as discussed above.
>
>
> #### Q3: Why were images in ImageNet-Dog resized to 96 x 96 x3?
> For a fair comparison, the ImageNet-Dog is resized to 96x96x3 following the previous methods, including DCCM[1] and DAC [2]. We have included the references about the image size in Sec.6 of the revised version.
>
>
> #### Q4: The number of experts and overclustering
> Thank you for your concern. In the cases where the number of the experts L differs from the number of ground-truth clusters K, the model will partition the datasets into L subsets instead of K. Even though the number of experts is currently tied with K, *it is not a drawback and does not prevent us from applying to the common clustering settings*. Also, MiCE does not use additional knowledge comparing to the baseline methods. If the ground-truth K is not known, we may treat K as a hyper-parameter and decide K following the methods described in [13][14], which is worth investigating in the future.
>
> The overclustering technique [10] is orthogonal to our methods and can be applied with minor adaptations. However, it may require additional hyper-parameter tuning to ensure overclustering improves the results. From the supplementary [11] provided by IIC [10], we see that the numbers of clusters (for overclustering) are set differently for different datasets. Despite overclustering is an interesting technique, we would like to **highlight the simplicity** of the current version of MiCE.
>
> We provide a possible implementation in the following for your further reference: To perform overclustering (of M clusters), we could add another (M+1) output layers and another gating and expert prototypes with M embeddings each. Therefore, on top of the original objective function with K experts, we can add another objective with M experts, where the possible values of the latent variables are in {1, 2, …, K} and {1, 2, …, M} respectively.
>
> We included the above discussion in Appendix F of the revision.

---

> > ### Author Response · Authors · 2020-11-18
> > **Response to Reviewer#2 (2/3)**
> >
> > #### Q5: Computational complexity
> > We find the training time of MiCE and MoCo to be comparable. It takes around 17 hours to train MoCo and 30 hours to train MiCE for 1000 epochs, respectively. We ran the experiments on a single GPU (NVIDIA GeForce GTX 1080 Ti). We have updated the submission to include the training time of MoCE and MiCE on CIFAR-10 in Appendix E.
> >
> > Comparing to MoCo, the additional training time of MiCE is mainly because of the mixture formulation and an extra forward pass of the mini-batch data. Please kindly refer to Appendix A.3 for the newly added Pytorch-like pseudocode for details.
> >
> >
> > #### Q6: Will $\boldsymbol{\mu}$ and $\boldsymbol{\omega}$ be similar, and their correspondence
> > We do not expect the two prototypes to be similar even though their dimensions are the same. In fact, they have different aims: the gating ones aim to divide the datasets into simpler subtasks for the experts, while the expert ones help the expert to solve the instance discrimination subtasks by introducing the class-level information. Therefore, the derived ELBO objective does not encourage the two sets of prototypes to be similar or maintaining a clear correspondence between them.
> >
> > Empirically, we calculate the cosine similarity between any pair of $\boldsymbol{\mu}$ and $\boldsymbol{\omega}$. The absolute values we get are less than 0.25, which showed that they are not similar. If we force them to be similar during training, the performance may be negatively affected due to the lack of flexibility.
> >
> > We added the discussion in Appendix F of the revised version.
> >
> >
> > #### Q7: Usage and effect of MMD
> >
> > Thank you for your concern. We use the centers of MMD as the gating prototypes that are always fixed during training and testing. To clarify the usage of MMD, we have provided the algorithm for generating the centers of MMD (Appendix B) and a Pytorch-like pseudocode of MiCE (Appendix A.3) in the revised version for your reference.
> >
> > As MMD provides the optimal inter-cluster dispersion, it can encourage embeddings of the gating network to be well-separated and can avoid the potential drawbacks when we have some of the gating prototypes crowded together. Specifically, compared to using gating prototypes that are samples from a uniform distribution, we find that using MMD can slightly improve the results by 1% ACC, as shown in Tab.3 of the original submission.
> >
> >
> > #### Q8: Max-pooling layer
> > Several contrastive learning methods [3][4][5] remove the first max-pooling layer when training on CIFAR-10 and CIFAR-100. We follow their implementations closely. Please kindly refer to [6][7] where we include the link to their code for your convenience. We ran an extra experiment with the max-pooling layer and get 83.6% ACC on CIFAR-10. The result is on par with the 83.5% we reported in Tab.1 of the revision. We have clarified it in Appendix D as well.

---

> > > ### Author Response · Authors · 2020-11-18
> > > **Response to Reviewer#2 (3/3)**
> > >
> > > #### Minor comment 1: What does NMI and ARI of > 1 mean? Don’t they have to be in the range of [0,1]?
> > > Yes, both have to be in the range of [0, 1]. In the experiment section, we show the percentage (%) of NMI, ACC, and ARI obtained by each of the methods. We have updated it in Sec. 6 of the revised version to clarify it. Thank you for your concern.
> > >
> > >
> > >
> > > *Lastly, we would like to express our **utmost appreciation for the constructive and positive comments** the reviewer provided again, which help us in improving the paper.*
> > >
> > >
> > >
> > >
> > > ====================================================================================
> > >
> > >
> > > [1] Jianlong Wu, Keyu Long, Fei Wang, Chen Qian, Cheng Li, Zhouchen Lin, and Hongbin Zha. Deep comprehensive correlation mining for image clustering. In Proceedings of the IEEE International Conference on Computer Vision, pp. 8150–8159, 2019.
> > >
> > > [2] Jianlong Chang, Lingfeng Wang, Gaofeng Meng, Shiming Xiang, and Chunhong Pan. Deep adaptive image clustering. In Proceedings of the IEEE international conference on computer vision, pp. 5879–5887, 2017.
> > >
> > > [3] Mang Ye, Xu Zhang, Pong C Yuen, and Shih-Fu Chang. Unsupervised embedding learning via invariant and spreading instance feature. In Proceedings of the IEEE Conference on Computer Vision and Pattern Recognition, pp. 6210–6219, 2019.
> > >
> > > [4] Zhirong Wu, Yuanjun Xiong, Stella X Yu, and Dahua Lin. Unsupervised feature learning via nonparametric instance discrimination. In Proceedings of the IEEE Conference on Computer Vision and Pattern Recognition, pp. 3733–3742, 2018.
> > >
> > > [5] Ting Chen, Simon Kornblith, Mohammad Norouzi, and Geoffrey Hinton. A simple framework for contrastive learning of visual representations. arXiv preprint arXiv:2002.05709, 2020.
> > >
> > > [6] https://github.com/mangye16/Unsupervised_Embedding_Learning
> > >
> > > [7] https://github.com/zhirongw/lemniscate.pytorch
> > >
> > > [8] Huang, Jiabo, et al. "Deep Semantic Clustering by Partition Confidence Maximisation." (2020): 8846-8855.
> > >
> > > [9] Wouter, Van Gansbeke, et al. "Learning To Classify Images Without Labels." (2020).
> > >
> > > [10] Xu Ji, Joao F Henriques, and Andrea Vedaldi. Invariant information clustering for unsupervised ˜ image classification and segmentation. In Proceedings of the IEEE International Conference on Computer Vision, pp. 9865–9874, 2019.
> > >
> > > [11] https://openaccess.thecvf.com/content_ICCV_2019/supplemental/Ji_Invariant_Information_Clustering_ICCV_2019_supplemental.pdf
> > >
> > > [12] Peter J. Rousseeuw (1987). “Silhouettes: a Graphical Aid to the Interpretation and Validation of Cluster Analysis”. Computational and Applied Mathematics 20: 53–65. doi:10.1016/0377-0427(87)90125-7.
> > >
> > > [13] McLachlan, Geoffrey J., and David Peel. Finite mixture models. John Wiley & Sons, 2004.
> > >
> > > [14] Smyth, Padhraic. "Model selection for probabilistic clustering using cross-validated likelihood." Statistics and computing 10.1 (2000): 63-72.

---

### Official Review · AnonReviewer3 · 2020-10-30
**Combination of existing ideas with concerns on technical details and experimental results**

**Rating:** 5
**Confidence:** 4

**Review:**

The paper proposes to use mixture of experts for image clustering.  The individual expert for each cluster adopts an instance discrimination approach for training.  The proposed method has shown superior clustering performance compared to an extensive number of clustering methods on a reasonable collection of data sets.

Though the overall idea of the proposed method is clear, the paper does not seem to explain some technical details clear enough.  In Equation, the probability P(Y,Z|X) does not seem to be correct. The product term over k appears to be valid only when it uses a hard assignment, i.e. P(z_n|x_n) = 1 for exactly one of the clusters.  The student and teacher embeddings are used before Equation 4 but have not been explained until a later part of the paper.  It is also unclear why an instance discrimination approach would lead to a better clustering performance.

Although an extensive number of clustering methods have been included in the experiments, it has omitted some strong competitors including Variational Deep Embedding (Jiang et al. 2016), Latent Tree Variational Autoencoder (Li et al. 2019), Deep clustering via a Gaussian mixture VAE with Graph embedding (Yang et al. 2019), etc.  Those methods appear to perform better on some of the data sets.  For example, those methods have been reported to yield over 85% of accuracy on the STL-10 data set but the proposed method yields only 75% on the same data set.  It would be better if the paper could include those methods in the experiments or justify the selection of baseline methods.

Overall, the proposed method appears to be an interesting combination of some existing methods.  However, the technical details need better clarity and the experiments should include more relevant methods.

---

> ### Author Response · Authors · 2020-11-18
> **Response to Reviewer#3 (1/2)**
>
> We thank the reviewer for the valuable and constructive comments and we have updated the submission accordingly. In particular, we clarified the main concerns on the technical details and selection of baselines. Please kindly find the detailed responses below.
>
>
> #### Q1: Concerns about the probability p(Y, Z|X)
> Thank you for pointing out the issue. The number of the equation seems to be missing in the comment the reviewer provided. As the reviewer mentioned $p(Y, Z|X)$, we suppose that the reviewer is referring to  Eq.(3) in the revision (which was Eq. (1) in the first/original submission) and will discuss it below. Please kindly let us know if the reviewer is referring to a different equation.
>
> There was a typo where the probability p(y_n, z_n=k | x_n=k) shall have an indicator function \mathds{1}(z_n = k) as its exponent. The correct Eq. (3) is:
> p(Y, Z |X) = \prod_{n=1}^N \prod_{k=1}^K p(y_n, z_n = k |x_n)^{ \mathds{1} (z_n = k)} = \prod_{n=1}^N \prod_{k=1}^K p(z_n = k | x_n )^{ \mathds{1} (z_n = k)} p(y_n | x_n, z_n = k)^{ \mathds{1} (z_n = k)}. The gating function p(z_n|x_n) is modeled as a categorical distribution and allows soft weighting on the predictions from the $z_n$-th expert. The formulation follows the Mixture of Experts (MoE) [1][2] closely.
>
> We have revised the submission accordingly. Please kindly refer to Eq.(3) in the revision for details. We also double-check the other equations and the algorithm, which remain correct.
>
> #### Q2: The student and teacher embeddings are used before Equation 4 but have not been explained until a later part of the paper.
> Thank you for the advice. We have added  *Preliminary* section to introduce contrastive learning methods, especially MoCo, before introducing MiCE to make the paper easier to follow. The technical details of MoCo, including the student and teacher embeddings, and the use of EMA are discussed in the Preliminary section. Please kindly refer to Sec.3 of the revision for more details.
>
> #### Q3: It is also unclear why an instance discrimination approach would lead to a better clustering performance
> As mentioned in the second paragraph of Sec.1 and the experiment section of the original submission, the instance discrimination task leads to discriminative representations [11], which are useful for clustering. In fact, improved representations can lead to better clustering results as shown in the clustering literature [3][4].
>
> Empirically, with instance discrimination, we can get a substantial improvement even with a simple clustering algorithm such as spherical k-means, as shown in Tab.1 in the revision. Further, with a unified framework, MiCE can improve the results by considering the semantic structure explicitly (also see Tab. 1).
>
> We added the above discussion in Sec. 1 of the revised version to make the motivation of using instance discrimination clearer.

---

> > ### Author Response · Authors · 2020-11-18
> > **Response to Reviewer#3 (2/2)**
> >
> > #### Q4: Omitting some of the prior works in experiments and need to justify the selection of baseline methods
> > We clarify that the experiment settings of the papers mentioned in the comments [5][6][7] and MiCE are **different in two aspects**. (1) We train the model **from scratch** without using a pre-trained model, following a large family of recent methods (IIC [8], MMDC [9], DCCM [10], etc.), while the papers the reviewer mentioned rely on pre-training. (2) We focus on a purely **unsupervised** setting. In contrast, VaDE [5] and DGG [7] use a **supervised pre-trained** model on ImageNet for STL-10. For fairness, in the original submission, we compared to many previous methods that use the same settings and did not include the papers the reviewer mentioned in the comment.
> >
> > In the revision, we have discussed the relation with papers the reviewer mentioned in Appendix D and clarified that we select baselines that are **without pre-training** in the first paragraph of Sec. 6.
> > Furthermore, we add preliminary experiments with **unsupervised pre-training** and MiCE obtains 89.3% accuracy on CIFAR10, which is much higher than 83.5% in Tab. 1 of the revision, showing that our proposal is orthogonal to the pre-training techniques. We will add more results and discussion in the camera-ready version.
> >
> > *Lastly, **we sincerely appreciate the reviewer for the insightful comments**, which indeed help us to improve the submission.*
> >
> >
> >
> >
> > ===================================================================================================================
> > [1] Robert A Jacobs, Michael I Jordan, Steven J Nowlan, and Geoffrey E Hinton. Adaptive mixtures of local experts. Neural Computation, 3(1):79–87, 1991.
> >
> > [2] Yuksel, Seniha Esen, Joseph N. Wilson, and Paul D. Gader. "Twenty years of mixture of experts." IEEE transactions on neural networks and learning systems 23.8 (2012): 1177-1193.
> >
> > [3] Chuang Niu, Jun Zhang, Ge Wang, and Jimin Liang. Gatcluster: Self-supervised gaussian-attention
> > network for image clustering. arXiv preprint arXiv:2002.11863, 2020.
> >
> > [4] Guy Shiran and Daphna Weinshall. Multi-modal deep clustering: Unsupervised partitioning of images. arXiv preprint arXiv:1912.02678, 2019.
> >
> > [5] Zhuxi Jiang, Yin Zheng, Huachun Tan, Bangsheng Tang, and Hanning Zhou. Variational deep embedding: An unsupervised and generative approach to clustering. arXiv preprint arXiv:1611.05148,
> > 2016.
> >
> > [6] Li, Xiaopeng, et al. "Learning latent superstructures in variational autoencoders for deep multidimensional clustering." arXiv preprint arXiv:1803.05206 (2018).
> >
> > [7] Yang, Linxiao, et al. "Deep clustering by gaussian mixture variational autoencoders with graph embedding." Proceedings of the IEEE International Conference on Computer Vision. 2019.
> >
> > [8] Xu Ji, Joao F Henriques, and Andrea Vedaldi. Invariant information clustering for unsupervised ˜ image classification and segmentation. In Proceedings of the IEEE International Conference on Computer Vision, pp. 9865–9874, 2019.
> >
> > [9] Guy Shiran and Daphna Weinshall. Multi-modal deep clustering: Unsupervised partitioning of images. arXiv preprint arXiv:1912.02678, 2019.
> >
> > [10] Jianlong Wu, Keyu Long, Fei Wang, Chen Qian, Cheng Li, Zhouchen Lin, and Hongbin Zha. Deep comprehensive correlation mining for image clustering. In Proceedings of the IEEE International Conference on Computer Vision, pp. 8150–8159, 2019.
> >
> > [11] Kaiming He, Haoqi Fan, Yuxin Wu, Saining Xie, and Ross Girshick. Momentum contrast for unsupervised visual representation learning. arXiv preprint arXiv:1911.05722, 2019.

---

### Public Comment · ~Wouter_Van_Gansbeke1 · 2020-11-10
**Missing relevant prior work**

Dear authors and reviewers,

We believe there is an important competing method omitted from both the related work section and the state-of-the-art comparison. In particular, the ECCV’20 paper ‘SCAN: Learning To Classify Images Without Labels’ is the current state-of-the-art in unsupervised image classification on CIFAR10, CIFAR100, STL-10, and ImageNet (1000 classes). Moreover the code of this paper was made publicly available, and can also be found on the relevant leaderboards of the well-known ‘Papers With Code’ platform (https://paperswithcode.com/task/unsupervised-image-classification).

Given these circumstances, we find it unfortunate that the state-of-the-art comparison failed to include the referred work. In particular, because the table does mention other works that are still under review, and report lower performance. We are looking forward to any future discussions.

---

> ### Author Response · Authors · 2020-11-18
> **Response to the public comment by Wouter Van Gansbeke**
>
> Thank you for the constructive comment. We find your paper inspiring and will include the comparisons with the proposed method in the revision.
>
> In the experiment section, we are mainly comparing to recent methods that do not involve pre-training. As far as we know, only DHOG [1] is still under review and all the other baseline methods are published. DHOG is included as it is also a relevant probabilistic clustering approach.
>
> We provide additional experiment results to compare with the method “SCAN” mentioned in the public comment fairly. As SCAN adopts a *three-step* training procedure, we think that it will provide additional insights by comparing MiCE/MoCo to SCAN at each step on CIFAR-10.
>
>
> #### 1st step:
> We find that the two-stage baseline results (MoCo+spherical $k$-means) are better than the SimCLR baseline results reported in your paper, despite MoCo and SimCLR performs similarly on ImageNet.  We suppose that using the spherical k-means as the output embeddings are L2-normalized, or setting the temperature to 1 may be able to improve the SimCLR baseline.
>
>
> #### 2nd step:
> **MiCE without pre-training outperforms SCAN with two steps of training**. SCAN obtains 78.7% and 81.8% on CIFAR-10 with the SimCLR augmentation and RandAugment respectively. In contrast, MiCE is able to get 83.4% despite we are using a weaker augmentation strategy following MoCo [3] and InstDisc [1].
>
>
> #### 3rd step:
> By fine-tuning the MiCE model with the same training protocol (except the learning rate can be smaller), *MiCE can obtain higher results comparing to SCAN*. After the self-labeling step, SCAN gets **10.0%** and **87.6%** ACC when training with the SimCLR augmentation and RandAugment respectively. While for MiCE, the results we obtain are as follows:
>
> 1.	MiCE pre-training + MiCE                                                      89.3%
> 2.	MiCE pre-training + MiCE (RandAugment)                        88.3%
> 3.	MiCE pre-training + MiCE (SimCLR augmentation)        **90.3%**
>
> To be specific, in the fine-tuning stage, we load the network parameters from the pre-trained model and use a smaller initial learning rate of 0.1 with all the other settings remain the same. We show the augmentation strategy in the parenthesis if we use a different one from the pre-training stage. For the RandAugment, we follow the implementation (and hyper-parameters) based on the link you provided. As mentioned in your paper, the self-labeling stage requires a shift in the augmentation, otherwise, it will lead to a degenerated solution. **In contrast, MiCE with pre-training is not prone to degeneracy and can get comparable or better performance than SCAN regardless of the choice of the augmentation strategy.**
>
> Besides, the self-labeling stage seems to be orthogonal to our approach and it can further improve MiCE as long as it is a general technique that is not sensitive to the difference in the previous stages.
>
> We included the above discussion in Appendix E of the revision.
>
>
> =========================================================================================================
>
> [1] Luke Nicholas Darlow and Amos Storkey. Dhog: Deep hierarchical object grouping. arXiv preprint arXiv:2003.08821, 2020.
>
> [2] Zhirong Wu, Yuanjun Xiong, Stella X Yu, and Dahua Lin. Unsupervised feature learning via nonparametric
> instance discrimination. In Proceedings of the IEEE Conference on Computer Vision
> and Pattern Recognition, pp. 3733–3742, 2018.
>
> [3] Kaiming He, Haoqi Fan, Yuxin Wu, Saining Xie, and Ross Girshick. Momentum contrast for unsupervised visual representation learning. arXiv preprint arXiv:1911.05722, 2019.

---

> > ### Comment · ~Wouter_Van_Gansbeke1 · 2021-06-08
> > **Reply to the authors: comparisons seem to be unfair**
> >
> > First of all, we would like to thank the authors for addressing our comments on the paper.  We really appreciate the effort by providing this short analysis. We believe that this analysis is insightful and a comparison with the current s-o-t-a is only fair.
> > Still, we would like to emphasize the following important points for the sake of being thorough. Our main concern is that the comparisons with prior s-o-t-a do not seem to be fair (i.e. architecture and train/test split) which makes most claims invalid.
> >
> > 1. SCAN [A] does not include the test set during training as opposed to the paper under review (see Table 3 in [A]). We acknowledge this is commonly the case in this field, but believe this is bad practice. If we include the test set during training, we also see a boost in accuracy of around 2-3%, so the numbers above seem plausible.
> >
> > 2. More importantly, we also noticed that the authors use a ResNet-34 when comparing with SCAN [A]. However, SCAN uses a ResNet-18. It is crucial to keep the comparisons fair. This can confuse the reader. Hence, the claim that MiCE outerforms SCAN is questionable.
> >
> > 3. If time permits, it would be interesting to see how well the proposed method performs on the other datasets, in particular on the complete ImageNet dataset (1000 classes) as is done for the first time in [A].
> >
> >
> > [A] Van Gansbeke, Wouter, et al. "Scan: Learning to classify images without labels." ECCV. 2020.
> >
> > (A late reply since ICLR did not allow public comments between 18 Nov. and the acceptance notification date.)

---

### Author Response · Authors · 2020-11-21
**Thank you for the valuable comments. Looking forward to further feedback.**

Dear Area Chairs and Reviewers,

We would like to thank the reviewers again for their constructive and insightful comments, which help us a lot in improving the submission. We have uploaded the revised version and responded to all the reviewers in detail. We believe that the quality of the paper is improved and the contributions are solid. In particular, we would like to highlight some key materials we added:

1.	Pytorch-like pseudocode of MiCE
2.	Several t-SNE visualizations of the embeddings learned by MiCE and MoCo
3.	Proof on the convergence of the EM algorithm under the approximation
4.	Algorithm on generating the expert prototypes $\boldsymbol{\omega}$ using MMD
5.	A preliminary section on contrastive learning to make the paper easier to follow

Besides, we also added extra experiments and discussions regarding the experiment settings, relevant baselines, and ablation studies in Appendix D, E, and F for your reference.

We understand that reviewers are busy during the response period, we would greatly appreciate it if the reviewers can kindly advise if our responses solve their concerns. If there are any other suggestions/questions, we will try our best to provide satisfactory answers.  We are looking forward to any further discussion with the reviewers. Thank you for your time.


Best regards,

The authors

---

### Decision · Program_Chairs · 2021-01-07
**Final Decision**

**Decision:**

Accept (Poster)

**Comment:**

Thanks for your submission to ICLR!

This paper considers a novel unsupervised image clustering framework based on a mixture of contrastive experts framework.  Most of the reviewers were overall positive about the paper.  On the positive side, they noted that the paper had an interesting idea, was well motivated, written well, and had solid results.  Also, the authors provided detailed and useful responses to the reviews, which further strengthened the case for accepting the paper.  On the negative side, one reviewer felt that the paper seemed a bit preliminary and its presentation could improve.  Also, there was some concern about missing comparisons / discussion to previous work (including from a public comment) or data sets (e.g. ImageNet-10).  Again, the authors responded well to these concerns.

Given that the overall response was quite positive with the paper, I'm happy to recommend accepting it.